# Locating macromolecular assemblies in cells by 2D template matching with cisTEM

Bronwyn A Lucas[1†], Benjamin A Himes[2†], Liang Xue[3,4], Timothy Grant[1], Julia Mahamid[3], Nikolaus Grigorieff[2]*

[1]Howard Hughes Medical Institute, Janelia Research Campus, Ashburn, United States; [2]Howard Hughes Medical Institute, RNA Therapeutics Institute, The University of Massachusetts Medical School, Worcester, United States; [3]Structural and Computational Biology Unit, European Molecular Biology Laboratory (EMBL), Heidelberg, Germany; [4]Collaboration for joint PhD degree between EMBL and Heidelberg University, Faculty of Biosciences, Heidelberg, Germany

**Abstract** For a more complete understanding of molecular mechanisms, it is important to study macromolecules and their assemblies in the broader context of the cell. This context can be visualized at nanometer resolution in three dimensions (3D) using electron cryo-tomography, which requires tilt series to be recorded and computationally aligned, currently limiting throughput. Additionally, the high-resolution signal preserved in the raw tomograms is currently limited by a number of technical difficulties, leading to an increased false-positive detection rate when using 3D template matching to find molecular complexes in tomograms. We have recently described a 2D template matching approach that addresses these issues by including high-resolution signal preserved in single-tilt images. A current limitation of this approach is the high computational cost that limits throughput. We describe here a GPU-accelerated implementation of 2D template matching in the image processing software *cis*TEM that allows for easy scaling and improves the accessibility of this approach. We apply 2D template matching to identify ribosomes in images of frozen-hydrated *Mycoplasma pneumoniae* cells with high precision and sensitivity, demonstrating that this is a versatile tool for in situ visual proteomics and in situ structure determination. We benchmark the results with 3D template matching of tomograms acquired on identical sample locations and identify strengths and weaknesses of both techniques, which offer complementary information about target localization and identity.

*For correspondence:
niko@grigorieff.org

†These authors contributed equally to this work

## Introduction

A major goal in structural biology is to understand the molecular mechanisms of biological processes that occur inside cells by studying the underlying proteins and their assemblies, collectively referred to here as 'complexes.' To realize this goal, X-ray crystallography and electron cryo-microscopy (cryo-EM) have been used to generate high-resolution density maps of purified complexes that could be interpreted by atomic models (*Berman et al., 2002*).

While the list of high-resolution structures is rapidly growing, a full understanding of molecular mechanisms requires the broader context of the cell (*Alberts, 1998*). This context is necessary to understand the functional coupling of different complexes, for example by transient interaction between them, by efficient shuttling of substrates and products, or by a common regulatory mechanism (*Braunger et al., 2018*; *O'Reilly et al., 2020*). Cryo-EM can be used to image complexes directly inside frozen-hydrated cells and tissues, making it one of the most promising approaches to add cellular context to structures that have been visualized in vitro. To date, the most developed

cryo-EM technique to visualize the 'molecular sociology' of cells (*Beck and Baumeister, 2016*) is electron cryo-tomography (cryo-ET) (*Oikonomou and Jensen, 2017*).

Similar to single-particle cryo-EM, cryo-ET has seen significant development over the past decade (*Wan and Briggs, 2016*), including the extension of subtomogram averaging to sub-nm resolution (*Schur et al., 2013*). Rather than recording a single view, cryo-ET involves collection and alignment of a series of images of an incrementally tilted specimen. The images are merged by reconstruction algorithms to form a tomogram, yielding 3D reconstructions of cells. However, the average resolution of a tomogram is inherently limited to ~20 Å (*Frank, 2006*) due to the finite angular sampling allowed by tolerable radiation damage, the increased specimen thickness at higher tilt angles, inaccuracy in the tilt series alignment caused by the weak signal present in individual tilted views, and local distortions in the sample during imaging (*Himes and Zhang, 2018*; *Voortman et al., 2014*). By adapting techniques from single-particle cryo-EM, subtomogram averaging can boost the resolution of reconstructions of complexes visualized in situ (*Bartesaghi et al., 2008*; *Winkler, 2007*). This approach has recently been extended to include the iterative refinement of tilt series alignment using a subtomogram average as a reference (*Himes and Zhang, 2018*; *Tegunov et al., 2021*).

Structure determination by subtomogram averaging relies on the initial detection of the complexes of interest in the raw tomogram (*Pfeffer and Mahamid, 2018*; *Zhang, 2019*). Semi-automated complex detection can be accomplished by matching with a 3D template (*Frangakis et al., 2002*), obtained by averaging a small set of manually selected subtomograms, or derived from another imaging modality, for example single-particle cryo-EM. 3D template matching (3DTM) has been particularly successful in the study of large cytoplasmic complexes, such as ribosomes, proteasomes and chaperonins (*Albert et al., 2017*; *Eibauer et al., 2012*; *Guo et al., 2018*; *Pfeffer and Mahamid, 2018*). However, this semi-automated detection requires extensive manual curation and the false positive rate is usually difficult to determine. Classification of subtomograms is required to both exclude false positives and to discriminate between structurally similar complexes (*Förster et al., 2008*, *Xu et al., 2012*). This process is computationally demanding and requires thousands of particles to yield well-defined class averages. As a consequence, it is practically impossible to reliably locate all but the most abundant complexes in tomograms. More generally, tomogram annotation depends critically on recognizing the overall shape, that is, the particle envelope, which may be obscured in regions of the cell with high molecular density, or in regions where several complexes connect to form a single continuous density (*Grünewald et al., 2002*). Methods that improve low-resolution contrast have been employed to aid in manual annotation, for example, imaging with a Volta phase plate (*Danev et al., 2017*) and using computational denoising, including classical linear and non-linear image filters and reconstruction algorithms (*Fernandez, 2012*) and more recently, deep-learning-based approaches (*Buchholz et al., 2019*). Although these strategies improve the sensitivity of detection in tomograms, they do not address the difficulty of detecting rare complexes or discriminating related structures in raw tomograms.

Recently, we described a 2D template matching (2DTM) technique that may overcome some of the limitations of 3DTM (*Rickgauer et al., 2017*). 2DTM matches projections of 3D templates to features found in single-exposure (2D) images of nominally untilted specimens, at 4 Å or higher resolution depending on the sample and targets to be localized (*Rickgauer et al., 2017*). Avoiding multiple exposures and high specimen tilt angles helps preserve the high-resolution signal in these 2D images (*Brilot et al., 2012*), and therefore, 2DTM can utilize this signal to detect complexes with high precision, as well as high angular and positional in-plane accuracy (x,y coordinates). In crowded environments such as the cell, this added signal comes at the expense of increased background (structural noise) in the images due to overlapping density from other molecules, and a relatively large error in localizing the depth of the targets within the sample (z coordinate) (*Rickgauer et al., 2017*). The matching of high-resolution details in 2DTM requires a fine-grained angular search composed of millions of reference projections and correlation maps to be calculated, making the computational workload of a 2D template search relatively high compared to a more coarse-grained search that is normally done with 3DTM. We describe here an implementation of 2DTM in the software package *cis*TEM (*Grant et al., 2018*), providing a user-friendly graphical interface and GPU-acceleration to speed up computation. We applied 2DTM to a set of images of *Mycoplasma pneumoniae* using a bacterial 50S large ribosomal subunit as a template. *M. pneumoniae* cells are small and do not require additional sample thinning (*Kühner et al., 2009*), for example via controlled cell lysis (*Fu et al., 2014*) or more commonly by using focused ion beam milling (*Marko et al., 2007*;

*Rigort et al., 2012*; *Strunk et al., 2012*). We benchmark the results of 2DTM in situ, using 3DTM to directly compare detection efficiency by collecting a subsequent tomogram of the same specimen area analyzed by 2DTM. We show that 2DTM is a versatile method, with comparable sensitivity and improved precision relative to 3DTM, pointing to potentially broad applications for both in situ visual proteomics and in situ structure determination, including de novo structure determination.

## Results

### GPU-accelerated 2DTM implemented in *cis*TEM

The large search space required for 2DTM makes this method computationally demanding; a single 1850 × 1850 pixel image required 1000 CPU-hours per search in the proof-of-principle MATLAB implementation (*Rickgauer et al., 2017*) and was improved about twofold by GPU-acceleration (*Rickgauer et al., 2020*) when comparing hardware of similar purchase price. To make 2DTM accessible to more users and a broader range of biological questions, we implemented 2DTM in *cis*TEM (*Grant et al., 2018*) using C++ (*Figure 1a*). This achieved a roughly 23x speed-up compared to the MATLAB CPU implementation, as measured on similar hardware, with measured times scaled to account for generational differences in CPU models. The core 2DTM algorithm is unchanged from its original description as depicted in the flowchart in *Figure 1—figure supplement 1*. The *cis*TEM implementation may be run as a standalone from the command line interface, or alternatively, using the *cis*TEM graphical user interface (GUI). In addition to a user-friendly interface, the GUI affords several advantages: Firstly, project metadata and image-specific information, such as CTF estimation, are tracked in a database; secondly, the search is easily divided over many CPUs or computers using *cis*TEM's MPI-like dispatch via run-profiles; and thirdly, the results are displayed in an interactive manner allowing for easy interpretation and comparison across multiple search conditions. A screenshot showing the results of a template matching search in the *cis*TEM GUI is shown in *Figure 1a*. The GUI displays the image searched (*Figure 1b*), maximum intensity projection (MIP) (*Figure 1c*) and the plotted results (*Figure 1d*), enabling rapid qualitative interrogation of the template matching results.

Even with these improvements, searching a more typically sized image, like that from a Gatan K3 detector (5760 × 4092 pixels) requires ~7100 CPU-hours when searching 13 focal planes, which are needed to determine the z coordinates of targets and resolve overlapping densities in a 150- to 200-nm-thick sample. Further substantial gains in the CPU-based code are not likely, given that about 85% of the computation for 2DTM is spent on calculating fast Fourier transforms (FFTs) using the already highly optimized Intel Math-Kernel library. To circumvent these limitations, we developed a GPU image class in *cis*TEM that has a subset of the same underlying member variables and methods as the corresponding CPU image class. Both implementations use positions on the Euler sphere to divide the search space (*Figure 2a*). The GPU implementation exploits further parallelism via threading, which combined with CUDA streams allows for multiple kernels to execute on the GPU simultaneously (*Figure 2b*). This in turn allows us to create a dynamic load balancing that results in full occupancy of the GPU over a wide range of problem sizes.

To evaluate the performance improvements of our GPU 2DTM implementation relative to the CPU 2DTM implementation, we compared two high-end GPUs (Nvidia GV100) against two high-end 28-core CPUs (Intel Xeon Platinum 8280) installed in the same general-purpose workstation, with all other hardware and variables unchanged. By this metric, the GPU-accelerated implementation of 2DTM achieved an 8.5x speed up relative to the CPU-only implementation (*Figure 2c*). Using IEEE 754 half-precision floating point values (FP16) for the arrays used to track search statistics, namely the pixel-wise sum and sum-of-squares over all orientations, resulted in further acceleration and a reduced memory footprint. The total speed-up was 10.5x (*Figure 2c*), relative to the CPU-only C++ implementation. The algorithm scales nearly linearly with the number of GPUs used, as shown for different NVIDIA GPU architectures in *Figure 2d*. The relative computational cost of each step of the GPU-accelerated 2DTM inner loop algorithm is detailed in *Figure 2—figure supplement 1*.

To avoid cumulative rounding errors at the reduced precision of FP16, we implemented a cascading summation where the sums and sum-of-squares are accumulated over ten search positions. Every tenth search, the results are accumulated in 32-bit single precision into a separate array. Every one-hundredth search, the results from the lower tier are accumulated, and so on, resulting in similar-

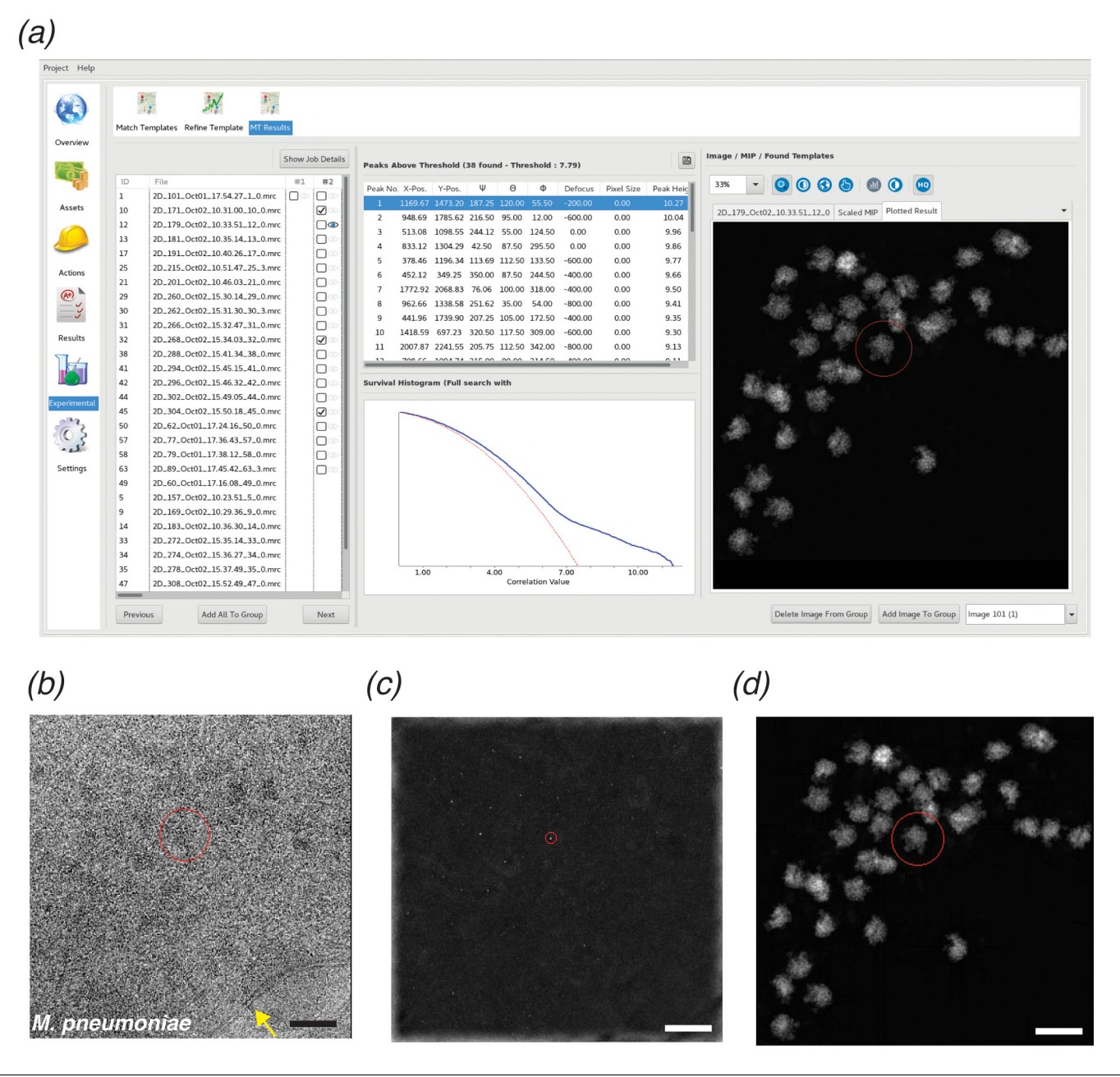

**Figure 1.** *cis*TEM GUI implementation of 2DTM. (a) Screenshot showing the results of a 2DTM search in the *cis*TEM GUI (located in the 'Experimental' tab). The panel on the left shows all images searched. Images may be searched individually (column #1) or as batch jobs (column #2). The Results tab shows the locations, orientations and SNR values of each detected target in a list, as well as the original image (b) (membrane highlighted in yellow), the maximum intensity projection (MIP, c) and the plotted result (d), which shows the best-matching orientation of the template at each detected location. The survival histogram (subpanel in (a)) shows the SNR values for all search locations (blue line) and compares this with the survival histogram of Gaussian noise (red line). This is used to establish the threshold at which a single false positive is expected per image. Scale bar in (b) = 500 Å. The online version of this article includes the following figure supplement(s) for figure 1:

**Figure supplement 1.** The 2DTM matching algorithm as implemented in *cis*TEM.

sized numbers being added. An additional consideration for the sum-of-squares array was needed as the smallest positive number that can be represented by FP16 without being rounded to zero is $2^{-14}$. To prevent these subnormal numbers from being flushed to zero, we temporarily multiply the array by a factor of $10^3$ until conversion to higher precision farther down the cascade.

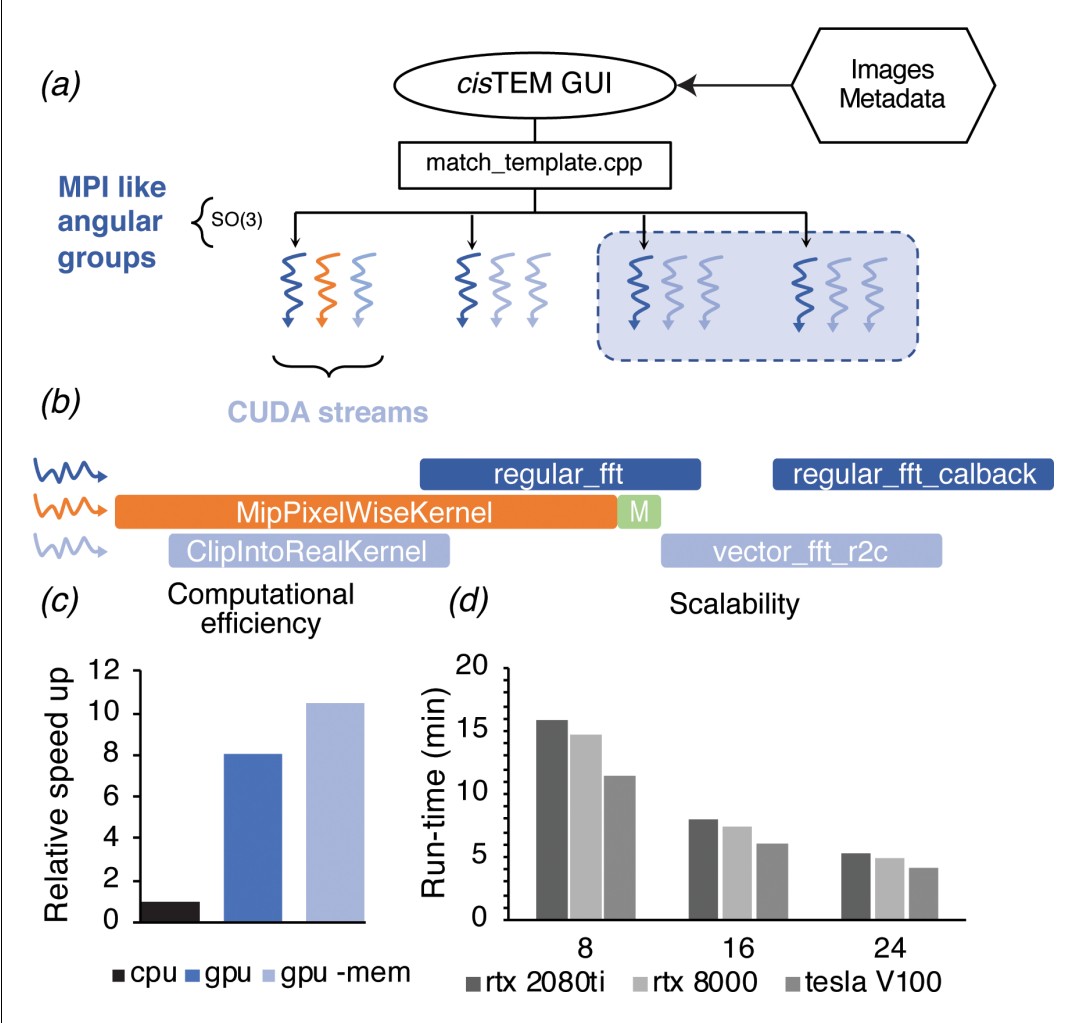

**Figure 2.** GPU acceleration of 2DTM in *cis*TEM. (**a**) The angular search space is distributed among any number of processors using the home-grown MPI-like socket communication in the *cis*TEM GUI (***Grant et al., 2018***). Unlike MPI, if fewer processors are available than requested (shaded box) processing may still proceed. (**b**) To expose further parallelism, additional host threads may be requested to subdivide each angular group to maximize occupancy on the GPU. Each host thread queues up a series of GPU kernels into its respective stream, and then returns to calculate the next projection and initiates its transfer to the GPU (green box). This way, close to 100% of the CPU and GPU is used during computation. (**c**) GPU acceleration relative to optimized CPU-based calculation, of which 85% is spent on MKL-based FFTs. Kernel-fusion using cuFFT callbacks and custom data structures combined with flexible kernel launch parameters ensure the GPUs stay saturated, enabling an 8x speedup. A total of 10.5x speedup is achieved by optimizing data throughput using the vectorized FP16 format for storing results. (**d**) The code scales nearly linearly with the number of GPUs and tracks with the total memory bandwidth of a given model. All timings were obtained using a padded K3 image with 4096 × 5832 pixels, and searching one defocus plane with 2.5°/1.5° angular steps.

The online version of this article includes the following figure supplement(s) for figure 2:

**Figure supplement 1.** GPU implementation and runtime profiling.

## Detection of 50S ribosomal subunits by 2DTM

The improved speed and increased throughput of 2DTM enabled us to perform an initial screen of template and search parameters that affect target detection in situ. To this end, we collected 2D images of plunge-frozen *M. pneumoniae, bacteria that* lack a cell wall and can be less than 200 nm thick, making them sufficiently thin to allow TEM imaging of whole cells in ice without the need for thinning (***Kühner et al., 2009***). We performed 2DTM using the *M. pneumoniae* 50S (PDB: 7OOD) (***Tegunov et al., 2021***) as a template (***Figure 3a***) and identified 6558 50S large ribosomal subunits in 220 2D images. This search did not distinguish between isolated subunits and subunits bound to the 30S small ribosomal subunit.

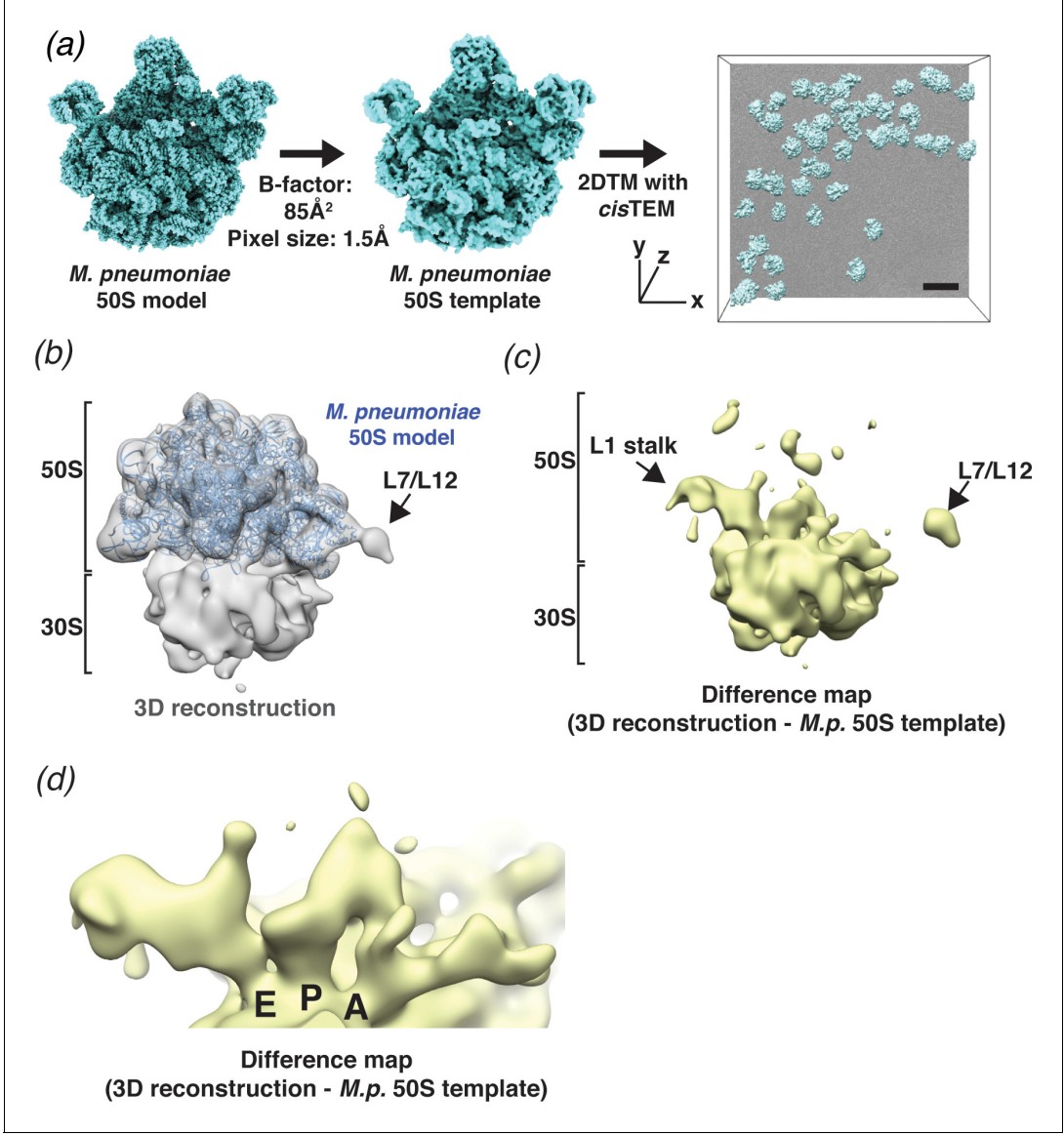

**Figure 3.** 2DTM detects ribosomes in *Mycoplasma pneumoniae* cells. (a) An overview of 2DTM: a cryoEM-like density was generated from an *M. pneumoniae* 50S model, a B-factor was applied and the resulting template used to identify locations and orientations of 50S in 2D images of *M. pneumoniae* cells with 2DTM implemented in *cis*TEM. Scale bar = 500 Å (b) 20 Å filtered 3D reconstruction generated using the locations and orientations of 5080 50S subunits detected in 220 images using 2DTM with a *M. pneumoniae* 50S template, showing clear density for the 30S ribosomal subunit (not included in the template). (c) Difference map showing the regions of the 3D reconstruction that differ from the 50S 2DTM search template. Arrows indicate additional density consistent with 70S ribosome structures. The difference map was generated with the same threshold as in (b). (d) A region of the difference map shown in (c), showing tRNAs in characteristic arrangements in the E, P, and A sites of the 30S subunit. *M.p.: M. pneumoniae*.

The online version of this article includes the following figure supplement(s) for figure 3:

**Figure supplement 1.** 2DTM results obtained using different settings.

The output from a 2DTM search are SNR values that depend on both the agreement between the template and the target as well as noise in the image (*Sigworth, 2004*). The noise is predominantly shot noise, as well as background generated by molecules and other cellular material overlapping the target in projection. A target is detected when the SNR value exceeds a threshold determined by the permitted number of false positives per search (usually one), as specified by the user (*Rickgauer et al., 2017*). To improve the match between template and recorded signal, the template can be low-pass filtered to approximate physical blurring of the image due to radiation

damage and beam-induced motion, as well as discrepancies between the target in the template, for example due to conformational differences. To low-pass filter the template, we applied a range of B-factors to the initial map generated using *pdb2mrc* (*Tang et al., 2007*, see Materials and methods) and found that the average SNR was maximized using a B-factor of about 85 $Å^2$ (*Figure 3—figure supplement 1a*). Using this value, we found that searching an image composed of exposure-weighted frames (32 e⁻/$Å^2$) (*Grant and Grigorieff, 2015*) increased the average SNR compared to using only the first eight frames (12.8 e⁻/$Å^2$) with or without exposure weighting (*Figure 3—figure supplement 1b*). We conclude that including additional frames with exposure weighting does not diminish the 2DTM SNR, indicating the constructive contribution of lower-resolution features to the detected signal. Consistent with SNR values being sensitive to defocus errors (*Rickgauer et al., 2020*; *Rickgauer et al., 2017*), searching each image with templates sampling a defocus range of 2400 Å in 200 Å steps (13 defocus planes) increased the number of detected targets by ~40% in a 100 nm thick sample and ~400% in a 220-nm-thick sample (*Figure 3—figure supplement 1c*). In addition to increasing the number of detected targets, the defocus search also provides a rough estimate of the out-of-plane position of each 50S. Earlier simulations predicted that protein background would result in reduced target detection in images collected at higher defocus (*Rickgauer et al., 2017*). In the present experiment, we did not observe a consistent relationship between mean SNR and image defocus over the range of ~500 to ~2200 nm underfocus (*Figure 3—figure supplement 1d*), suggesting that detection of complexes as large as 50S ribosomal subunits is not prevented by collecting images farther from focus.

## 3D reconstructions of complexes localized by 2DTM

In our current 2DTM implementation, each detected target is assigned an x,y location, three Euler angles and a defocus value (z coordinate) (*Figure 3a*). These parameters can be used to calculate a 3D reconstruction using standard single-particle methods. It is well known that a reconstruction calculated from particles that were identified using a template may suffer from significant template bias (*Henderson, 2013*), reproducing only features of the template and not the targets. However, as shown by *Rickgauer et al., 2017* and discussed below, applying an absolute (rather than relative) threshold based on a known noise distribution, also limits template bias and new features not present in the template may be visible in a reconstruction derived from detected targets. To test this, we calculated a reconstruction using the results from searching 220 images of *M. pneumoniae* cells (see above), selecting targets only from the best images that had more than nine detected targets and at least one target with an SNR value above 9. Using the 5080 targets that met these criteria, out of the total of 6558 detected targets, we calculated a 3D reconstruction (*Figure 3b*). The reconstruction had a nominal resolution of 4.3 Å (FSC threshold of 0.143, *Rosenthal and Henderson, 2003*; *Figure 3—figure supplement 1e*). However, visible density of known ribosome features (not shown) suggested that this resolution estimate was unrealistically high, as expected due to the well-known effect of template bias (*Stewart and Grigorieff, 2004*). We used the masked FSC curve between the *M. pneumoniae* 30S structure, which was not included in the template search, and the 3D reconstruction to define an appropriate resolution to filter the reconstruction (see Materials and methods, *Figure 4—figure supplement 1d*). The 20 Å-filtered reconstruction reproduces the 50S template as expected, but also shows clear, albeit weaker density for the 30S subunit that was not present in the template, and thus cannot be due to template bias (*Figure 3b,c*).

The difference map calculated using *diffmap* (*Grigorieff, 2021a*), between the *M. pneumoniae* 50S template and the 3D reconstruction shows density in regions of the 50S that have been shown to be flexible from in vitro reconstructions of the ribosome, specifically around the L1 stalk and the L7/L12 stalk (*Figure 3c*). The *Escherichia coli* L1 stalk moves ~45–60 Å relative to 50S during translocation (reviewed in *Ling and Ermolenko, 2016*), and the C-terminal domain of L10 and N-terminal domain of L12 are known to be flexible (*Diaconu et al., 2005*). Moreover, we observed density in each of the three tRNA-binding sites on the small subunit, which were not present in the template (*Figure 3d*). The density is consistent with tRNAs representing multiple states and may also reflect the binding of other factors, such as translational GTPases in the A-site. It is therefore likely that the calculated reconstruction represents an average of different conformations adopted by the ribosome in vivo, as expected in actively growing cells. This average differs therefore from the single conformation used for the template, giving rise to the features observed in the difference map. Smaller features in the difference map may also correspond to noise in the reconstruction, as well as features

that result from inaccuracies in the resolution-dependent scaling of the template against the reconstruction before subtraction. More accurate scaling may be achieved by scaling according to local resolution estimates, which were not obtained here, as well as masking to exclude non-overlapping parts in both maps.

## 2DTM reveals species-specific structural features

To test the precision of 2DTM and to evaluate whether species-specific structural differences would preclude detection by 2DTM, we searched the same 220 images of *M. pneumoniae* with a 50S template derived from *Bacillus subtilis* (PDB: 3J9W). *B. subtilis* 50S is structurally closely related to *M. pneumoniae* 50S but displays differences in rRNA sequence and protein composition (*Grosjean et al., 2014*; *Sohmen et al., 2015*; *Figure 4a*). Comparing the two models by FSC and taking 0.5 as a similarity threshold (*Figure 4—figure supplement 1a*) suggests that the effective resolution of the *B. subtilis* template is limited to ~4.7 Å. Using *B. subtilis* 50S as a template, we identified 2874 50S locations, less than half than were identified using the *M. pneumoniae* template (*Figure 4b*), and with significantly lower 2DTM SNR values (p<0.0001, Mann-Whitney U test, *Figure 4c*). Again, to limit targets to the best images, we included only images with more than two detected targets and at least one target with an SNR value above 9. Using the 1172 targets from these images, we calculated a 3D reconstruction (*Figure 4d*). As before, the 20 Å-filtered reconstruction reproduces the 50S template with clear density for the 30S subunit and shows features consistent with translating ribosomes (*Figure 4d–e*). However, unlike before, the 3D reconstruction also shows extensive differences in the 50S subunit relative to the *B. subtilis* 50S template (*Figure 4e*). Since this reconstruction was generated using ~fivefold fewer particles, some of the additional density likely reflects an expected higher level of noise. Despite this, the 3D reconstruction revealed that some of the features visible in the 50S density deviate from the *B. subtilis* template and likely result from *M. pneumoniae*-specific features (*O'Reilly et al., 2020*; *Figure 4e*, *Figure 4—figure supplement 1c*). Specifically, the 3D reconstruction showed density consistent with *M. pneumoniae*-specific C-terminal extensions of proteins L29 and L22, and protein L9 (*Figure 4f*, *Figure 4—figure supplement 1c*; *O'Reilly et al., 2020*). The latter was absent from the *B. subtilis* model despite being encoded in the *B. subtilis* genome (*Sohmen et al., 2015*). The 3D reconstruction also lacked density for protein uL30, which is present in the *B. subtilis* template, but absent from the *M. pneumoniae* genome (*Grosjean et al., 2014*; *Figure 4g*). Moreover, the 3D reconstruction shared additional unattributed features with the previously determined reconstruction (*Figure 3b*) that do not derive from either template (*Figure 4—figure supplement 1c*). We conclude that a *B. subtilis* 50S template can be used as a homology model to identify *M. pneumoniae* ribosomes and distinguish species-specific features, and that including high-resolution signal in 2DTM does not overly bias the 3D reconstruction when an appropriately high SNR threshold is used. These results further demonstrate the reliability of 2DTM and its potential to directly resolve complex structure in situ.

## Benchmarking of 50S ribosomal subunit detection by 3DTM

Cryo-ET combined with 3DTM is currently one of the most commonly used approaches for locating molecules within cells using available structural information. Rather than using high-resolution templates as is done in 2DTM, 3DTM locates molecules in tomograms with templates filtered to 30 Å or lower (*Himes and Zhang, 2018*). To compare the detection of 50S ribosomal subunits by 2DTM and 3DTM, we collected 19 2D images of *M. pneumoniae* cells followed by tomograms of an overlapping area. These tomograms were collected after the 32 e⁻/Å² exposure for images used with 2DTM, using standard protocols (*Figure 5a*). The pre-exposure of 32 e⁻/Å² affects the signal in the tomograms at higher resolution but is expected to have only a small effect in the resolution range relevant for 3DTM, that is, 20 Å and lower (*Grant and Grigorieff, 2015*). Consistent with this prediction, we estimated an effective B-factor for the subtomograms of 2000 Å² using *RELION*, compared to 1700 Å² without pre-irradiation (*O'Reilly et al., 2020*). The difference of 300 Å² attenuates the signal at 30 Å by only an additional 8%. Furthermore, the large B-factors in both cases confirm the loss of high-resolution signal in the raw tomograms (see Introduction), which currently limits 3DTM to low-resolution.

To identify 50S subunits with 3DTM, we followed established protocols using PyTom and a 30 Å low-pass filtered 50S template generated in a previous study (*O'Reilly et al., 2020*). The found

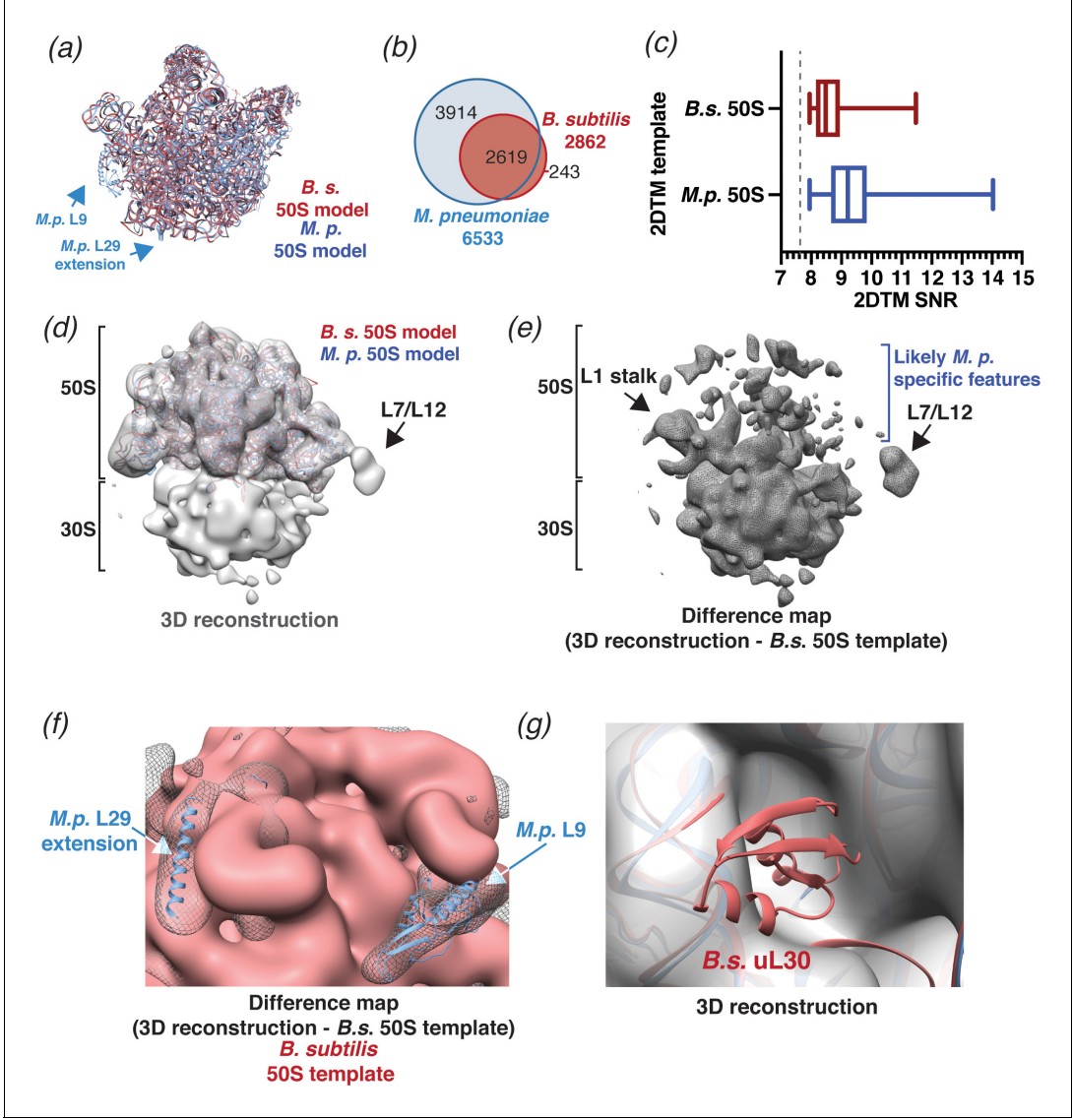

**Figure 4.** 2DTM using a *B. subtilis* 50S template reveals species-specific structures. (a) Molecular models of *B. subtilis* (red) and *M. pneumoniae* (blue) 50S ribosomal subunits aligned using *UCSF Chimera* (**Pettersen et al., 2004**). (b) Venn diagram showing the number of 50S subunits detected in the same dataset of 220 images of *M. pneumoniae* cells using the indicated template. (c) Boxplot showing the distribution of 2DTM SNR values of the locations quantified in the diagram in (b). The width of the box indicates the interquartile range, the middle line indicates the median and the whiskers indicate the range. The dashed vertical line indicates the 2DTM SNR threshold used. (d) 20 Å filtered 3D reconstruction generated using the locations and orientations of 1172 50S subunits detected in 220 images using 2DTM with a *B. subtilis* 50S template, showing clear density for the 30S ribosomal subunit and L7/L12 (not included in the template). The threshold was selected to reflect the threshold used in *Figure 3b and c*. (e) Difference map showing the regions of the 3D reconstructions that differ from the 50S 2DTM search template. Arrows indicate additional density consistent with 70S ribosome structures. (f) Difference map as described in (e) (gray mesh), aligned to the *B. subtilis* 50S template (red) and both *M. pneumoniae* (*M.p.*, blue) and *B. subtilis* (red, not visible) molecular models. The difference map is shown at the same threshold as in (d). (g) 3D reconstruction as described in (d) (transparent gray), aligned to *M. pneumoniae* (*M.p.*, blue) and *B. subtilis* (*B.s.*, red) molecular models.

The online version of this article includes the following figure supplement(s) for figure 4:

**Figure supplement 1.** Comparison of 2DTM results using different 50S templates.

targets were ranked according to their cross-correlation score and the top 600 hits were selected for each tomogram, followed by alignment and 3D classification with *RELION* 3.0.8 (*Zivanov et al., 2018*). The selection of the top 600 hits ensured that more than 90% of potential targets in one tomogram, including free 50S and 50S in assembled 70S, were included. Combining *RELION* classification results with cross-correlation ranking (*Figure 5—figure supplement 1*) shows that the hits

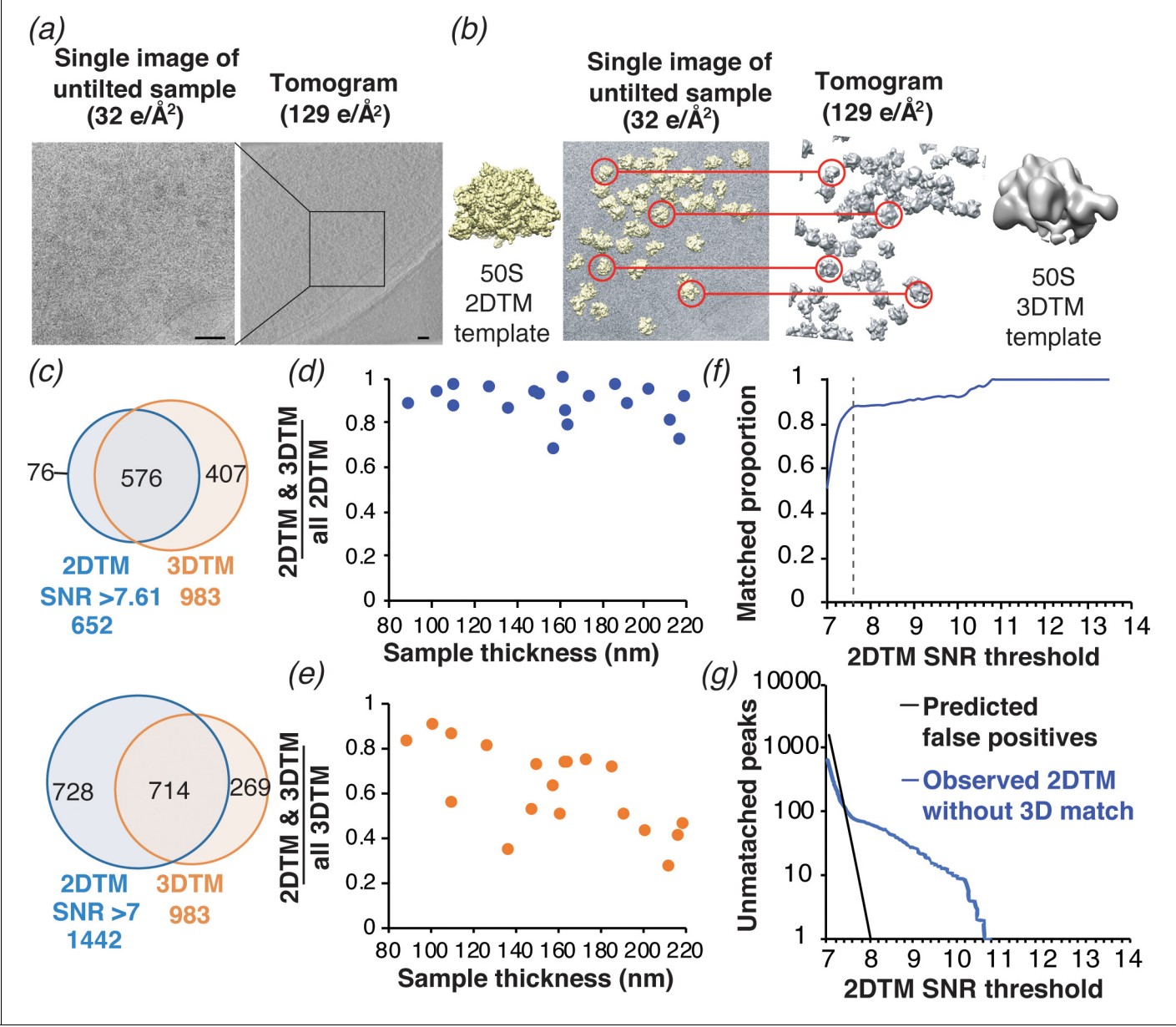

**Figure 5.** Comparison of ribosome detection by 2DTM and 3DTM. (a) Images of untilted cryo-EM grids of *M. pneumoniae* were collected with a total exposure of 32 e⁻/Å², followed by a tilt series of an overlapping region with a total exposure of 129 e⁻/Å² to reconstruct a tomogram. (b) 50S ribosomal subunits were identified in the 2D images by 2DTM with the *M. pneumoniae* 50S (left) and in the 3D tomogram by 3DTM using the 50S subunit as a template (right). The 2DTM and 3DTM templates were aligned to ensure that the respective coordinate systems were aligned, and that the x,y coordinates of the detected 50S subunits in each search could be aligned. (c) The proportion of 2DTM and 3DTM coordinates within 100 Å and 20° in each of the three Euler angles in 19 images was calculated using an SNR threshold that allowed either one false positive per image (upper), or detection of ~2 times more potential 50S targets (lower). (d) Plot showing the proportion of 2DTM targets that were also detected by 3DTM as a function of sample thickness. (e) Plot showing the proportion of 3DTM targets that were also detected by 2DTM as a function of sample thickness. (f) Plot showing the proportion of 2DTM 50S targets with a positional and rotational 3DTM match at the indicated 2DTM SNR threshold (dashed line). (g) Plot showing the number of expected false positives in the 2DTM search assuming a Gaussian noise model (black) and the observed number of 2DTM targets without a matching 3DTM target (blue) at the indicated 2DTM SNR.

The online version of this article includes the following figure supplement(s) for figure 5:

**Figure supplement 1.** Analysis of targets detected by 3DTM.

**Figure supplement 2.** Comparison of 2DTM and 3DTM results.

with the lowest scores contained fewer than 10% of 50S and 70S targets. The proportion of false positives was consistent with a previous study (*O'Reilly et al., 2020*), indicating that pre-irradiation did not noticeably affect 50S or 70S detection by 3DTM.

We aligned the coordinates of the 652 50S subunits identified by 2DTM with 983 50S subunits identified by 3DTM (see Materials and methods), both containing 50S targets corresponding to single 50S subunits and 70S ribosomes (*Figure 5b*). Within the area common to the 2D images and tomograms, we identified 576 2DTM targets that were within a 100 Å distance in the x,y plane, and within 20° in each Euler angle of a 3DTM target. These limits included ~95% of all paired targets (*Figure 5c–d*). The paired targets represent ~90% of all 2DTM targets and ~60% of all 3DTM targets in this area (*Figure 5c*, upper). We found that the proportion of 2DTM targets with a corresponding 3DTM target was consistent across the images examined (*Figure 5d*). In contrast, the proportion of the 3DTM targets detected in the 2DTM search was variable and showed a negative correlation with sample thickness (*Figure 5e*). This agrees with our and prior observations that 2DTM is particularly sensitive to sample thickness (*Figure 5—figure supplement 2*; *Rickgauer et al., 2020*; *Rickgauer et al., 2017*), which likely contributes to the high false negative rate of 2DTM relative to 3DTM.

The number of targets detected in 2DTM depends on the SNR threshold, which is determined by the desired false-positive rate. Assuming a Gaussian noise distribution, we calculated a threshold SNR of 7.61 to allow for a false positive rate of one per image (*Rickgauer et al., 2017*). To test if a lower SNR threshold improves the agreement between 2DTM and 3DTM, we determined the proportion of detected 2DTM 50S targets with a matched 3DTM target at different SNR thresholds (*Figure 5f*). Decreasing the SNR threshold to 7.00 increased the number of detected targets more than twofold (from 652 to 1442), but the number of 2DTM targets with a corresponding 3DTM target increased only ~1.2 fold (from 576 to 714) (*Figure 5c*, lower). The plot of matching targets in *Figure 5f* shows that at SNR thresholds > 7.6, the proportion of 2DTM/3DTM paired targets was ~90–100%, while lowering the SNR threshold below 7.6 resulted in a sharp decrease, indicating that below this threshold, more non-matching targets were found than matching targets, many of them likely false positives. The value of 7.6 agrees closely with the 7.61 threshold at which one false positive per image is expected (*Figure 5f*, dashed line). This experimentally validates our use of a Gaussian noise model and the Neyman-Pearson threshold criterion and shows that the SNR threshold can be used to set a desired false positive rate suitable for a particular experimental design.

At a threshold of 7.61, the number of 2DTM targets not detected in the 3DTM search was 76, > 4 times higher than the expected 19 false positives for the 19 searched images and representing ~12% of the 2DTM peaks (*Figure 5g*). This is consistent with the estimated 3DTM false negative rate of ~10% (*Figure 5—figure supplement 1*), and suggests that these indeed represent false negatives in the 3DTM search. We also noted that in samples of ~100 nm thickness we detected ~90% of the 50S subunits identified with 3DTM (*Figure 5e*). We therefore estimate that given optimum sample parameters, 2DTM can detect 50S subunits with comparable sensitivity to 3DTM.

## 2DTM can validate targets identified by 3DTM

While 2DTM tends to miss targets close to the chosen SNR threshold, one of the key limitations of 3DTM is a high false discovery rate and the difficulty in discriminating between true and false positives. This shortcoming can be partially mitigated by manually curating the detected 3DTM targets and subsequent classification of the subtomograms. The manual curation requires experience, is time-consuming and generally not reproducible. Progress has been made to improve the discrimination between true and false positives using deep learning algorithms as demonstrated with synthetic data (*Gubins et al., 2019*), although on real data they currently perform only as well as 3DTM (*Moebel et al., 2020*).

We investigated the use of 2DTM as an independent screen to validate 3DTM targets, by comparing the list of detected 3DTM targets before manual curation and 3D classification with the list of high-confidence 2DTM targets. We found that 2DTM detects ~60% of the 3DTM targets that are classified as 50S or 70S, while only detecting <5% of the 3DTM targets that likely represent false positives (*Figure 6a*). Thus 2DTM can clearly discriminate targets that were excluded by classification of 3DTM targets with *RELION* (*Bharat and Scheres, 2016*). This demonstrates that 2DTM, unlike 3DTM, can directly and reliably locate targets in situ without the need for additional refinement or

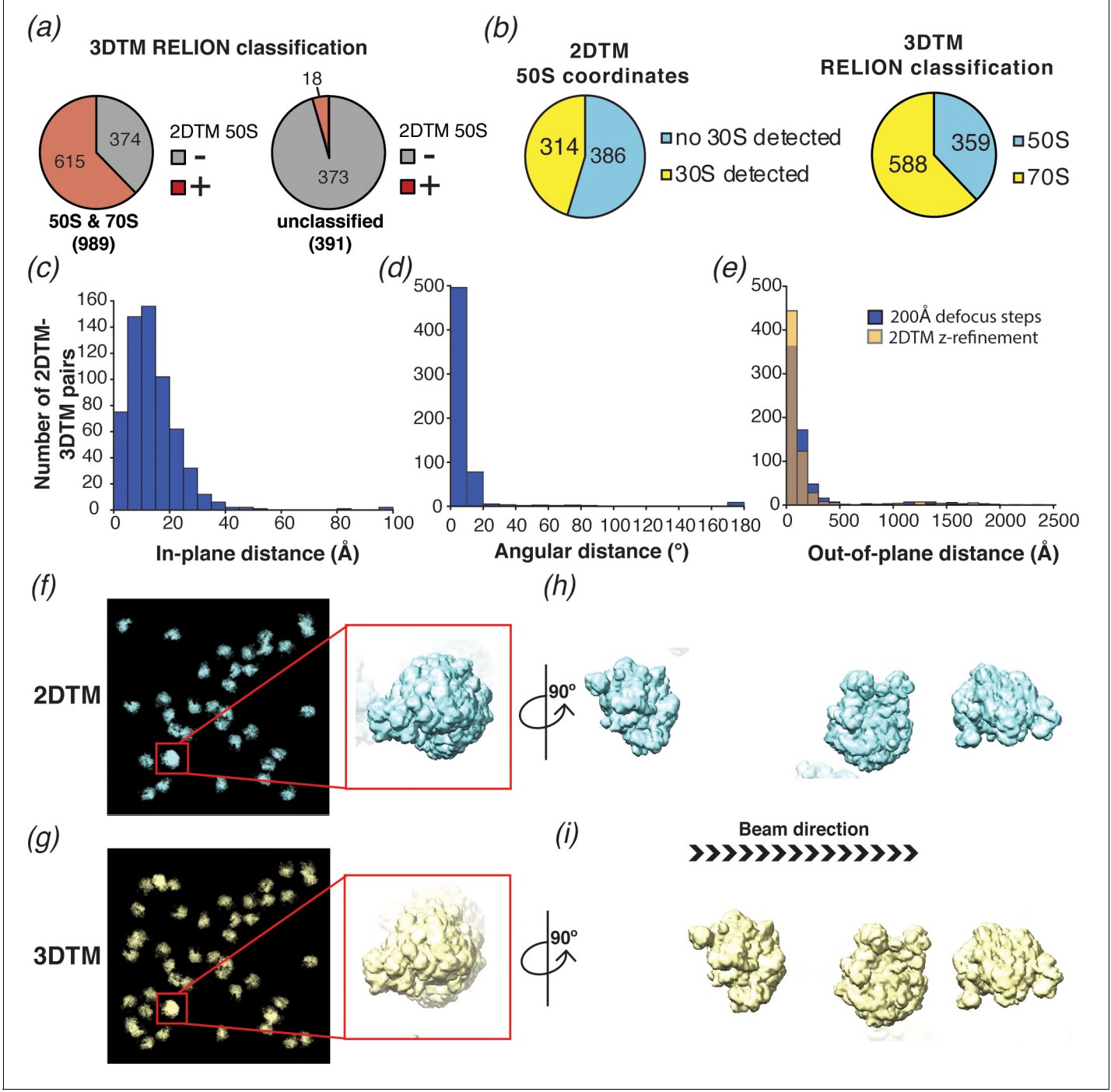

**Figure 6.** 2DTM is precise, excludes non-ribosome particles and permits detection of ribosomes overlapping along the projection direction. (a) Pie chart showing the results of a comparison of a set of 1380 3DTM coordinates initially identified by PyTom followed by 3D classification in *RELION* that identified 989 targets as 50S or 70S (left) and 391 targets as non-ribosomal particles (right), with a list of 652 50S 2DTM targets. Red indicates the proportion aligning with 50S coordinates identified by 2DTM; gray indicates non-matching 3DTM coordinates. (b) Pie charts showing the proportion of 50S detected by 2DTM with the *M. pneumoniae* template with a bound 30S target as determined by performing a local search with *refine_template* (see Materials and methods) (left), and the ratio of 3DTM 50S targets classified as 70S or 50S by *RELION* (right). (c) Histogram showing the distribution of the in-plane distance between matched 3DTM and 2DTM after 3DTM refinement of the subtomograms with *RELION*. (d) As in (c), showing the angular distance. (e) As in (c), showing the out-of-plane difference (z coordinate) before (blue) or after (yellow) 2DTM refinement of z coordinates. (f) Plotted result from a 2DTM search showing template projections at the locations and Euler angles of detected 50S subunits (left), inset showing two overlapping 50S subunits when viewed parallel to the image plane. (g) As in (f), showing the plotted result from 3DTM of the same area aligned to

*Figure 6 continued on next page*

*Figure 6 continued*

show the same perspective. (h) The template projections in (f) rotated 90° to show the overlapping 50S subunits perpendicular to the image plane. (i) As in (h), showing the result from 3DTM in the same area.

The online version of this article includes the following figure supplement(s) for figure 6:

**Figure supplement 1.** Proportional bar chart showing the percentage of all 983.

curation. It also shows that 2DTM provides complementary evidence that could be used to validate targets identified by 3DTM without the need for subtomogram averaging and classification.

To evaluate whether we could locate 30S subunits associated with 50S by template matching, and to compare the results to the *RELION* classification of the 3DTM results as 50S or 70S, we used the locations and orientations of the 50S targets detected with the *M. pneumoniae* template to perform a local search for the 30S subunit (PDB: 7OOC) with and without the flexible head domain (see Methods), sampling a range of ±12.5 degrees, that is, larger than the rotational range of the 30S relative to the 50S. Of the 700 50S locations searched, 314 (~45%) had detectable 30S (*Figure 6b*), fewer than the ~70% of the 3DTM targets that were classified as 70S using *RELION*. We speculate that this is due to structural discrepancies that likely exist between the 30S template and specific instances of the small ribosomal subunit in our images. The 30S subunit is known to undergo significant conformational changes during the functional cycle of the ribosome in addition to inter-subunit rotation (see Discussion).

We also compared the x,y locations (*Figure 6c*), z coordinates (*Figure 6e*), and orientations (*Figure 6d*) assigned by 2DTM to those assigned by 3DTM. Comparing the 2DTM results with the refined 3DTM results shows that ~ 95% (576 out of 603) of the targets aligned in x and y had an angular distance of 20° or less, with a median of 7° (using *Equation 1*; *Figure 6d*). The median in-plane distance of 12 Å is likely due to deformation of the sample under the electron beam, which is more pronounced with the higher electron dose used in tilt series, and to a lesser extent due to the lower resolution of 3DTM. In contrast to the x,y locations, which were calculated in steps of one pixel (1.27 Å), the defocus values of 2DTM were sampled in 200 Å steps and, correspondingly the median out-of-plane distance between detected 2DTM and 3DTM targets was much higher at ±84 Å (*Figure 6e*). We used the program *refine_template* (see Materials and methods) to perform a local defocus refinement, which reduced the median out-of-plane distance to ±59 Å (*Figure 6e*), less than 1/3 the diameter of the 50S. The fivefold greater error in z relative to the x,y plane is consistent with the previously reported weaker dependence of the 2DTM SNR values on defocus (*Rickgauer et al., 2017*).

We note that some of the 50S subunits detected by 2DTM overlap in the image and were found in comparable locations by 3DTM (e.g. *Figure 6f–i*). This confirms that 2DTM, as previously proposed (*Rickgauer et al., 2017*) can effectively detect overlapping particles without sample tilt.

## Discussion

In this study, we provide strategies to overcome some of the major limitations of molecular localization in cells using 2DTM. Firstly, we present a new implementation of 2DTM in *cis*TEM, which offers a user-friendly interface and substantial acceleration by running the most demanding computations on GPUs. Secondly, we show that averaging of complexes identified with 2DTM can reveal differences between the template and the target and allows identification of interacting complexes that may be difficult to detect alone. Thirdly, we demonstrate that 2DTM detects bona fide ribosomes in *M. pneumoniae* cells with high precision and comparable sensitivity relative to 3DTM. In the following, we highlight some of the technical advances of our implementation, discuss applications of, and possible improvements to 2DTM, and how 2DTM and 3DTM can benefit from each other.

### GPU acceleration increases the throughput of 2DTM

To make 2DTM useful in practice, we have increased its speed by more than 10-fold, socket-to-socket, relative to our C++ CPU implementation in *cis*TEM via GPU acceleration, making it possible to search multiple images in hours instead of days. To fully utilize the GPU capacity, we have introduced flexible load balancing via CUDA streams, accessible to the user by including additional CPU

threads in their *cis*TEM run profile. This ensures a simple mechanism to take full advantage of computers containing variable Nvidia GPU hardware architectures and adapt to different problem sizes.

GPUs have so many processing units (cores) that efficient algorithms, like the FFT, are often limited by memory transfers. We have ameliorated this problem by taking advantage of the FP16 format, storing two variables in the low and high 16 bits of a 32 bit __half2 vector data type, thereby roughly doubling the memory bandwidth. Due to the low SNR intrinsic to cryo-EM data, the loss of numeric precision associated with FP16 does not substantially alter the effective SNR of the data stored in FP16 format. We expect that this could be useful for computation in other modalities of cryo-EM.

With a few precautions, we can retain sufficient precision, that is, greater than what is used for our detection threshold, to obtain the same net results as with the 64-bit double precision accumulators used in the CPU code. We have also taken advantage of increased memory bandwidth of FP16 by converting the FFT of the input image, which is only transformed once at the outset of the algorithm, into FP16 format. When this array is read into the GPU's streaming multi-processors for cross-correlation, it is converted to 32-bit single precision as part of our cuFFT callback routine prior to the conjugate multiplication and inverse FFT. Due to the limited range of FP16, we did not investigate whether this reduced precision could be used directly in the FFT in 2DTM. In principle, this could yield another ~1.6x speedup by doubling the inverse FFT computation speed. Additionally, it may be possible to exploit the fact that each template is padded with zeros to the full image size prior to taking the FFT, resulting in many redundant zero-valued 1D FFTs in the common row-column transform approach for 2D FFTs.

Support for the new Bfloat16 floating-point format, which has a larger range than FP16, and hardware acceleration for asynchronous memory movement on chip in the new Nvidia Ampere architecture, may also provide additional opportunities to accelerate the current algorithm, which is predominantly memory-bandwidth limited. Finally, further acceleration may be gained by improving the algorithm itself, perhaps via a hierarchical search as is done in single-particle cryo-EM. For example, a coarse search followed by a local refinement may improve both the speed and overall accuracy of 2DTM.

## Template matching and noise overfitting

Template matching is a well-established method to pick particles in single-particle cryo-EM images for further processing and 3D reconstruction (*Scheres, 2015*; *Sigworth et al., 2010*). The advantage of using a template that matches the structure of the particle to be reconstructed is more discriminatory picking, and the possibility to exclude contaminants and other features in an image that do not represent valid particles. Using a template for picking can also lead to template bias, that is, a 3D reconstruction that reproduces the template rather than the true particle structure (*Henderson, 2013*; *Subramaniam, 2013*; *van Heel, 2013*). It is therefore important to validate features in a reconstruction derived from particles identified by template matching. This is commonly done by observing features in the reconstruction that were not present in the template, and that are known to be true. For example, templates can be low-pass filtered to 20 Å or lower (*Scheres and Chen, 2012*), and the emergence of high-resolution features such as secondary structure or amino acid side chains will then validate the reconstruction. In the current release of *cis*TEM (*Grant et al., 2018*), a Gaussian blob is used for particle picking, and recognizable low- and high-resolution features visible in the reconstruction serve as validation. Since 2DTM uses high-resolution features to identify targets, reconstructions have to be validated differently, for example by identifying additional a priori-known density features that were not present in the template. In the case of using the 50S templates for 2DTM, the reconstructions showed clear density for the 30S subunit, as well as at tRNA binding sites, both of which validate the reconstructions (*Figure 3c–d*). However, despite this validation, there may still be a template bias in the reconstruction (*Stewart and Grigorieff, 2004*). While it is not possible to quantify this bias without knowledge of the unbiased reconstruction, the strength of the bias will depend on the number of degrees of freedom accessible during template matching, that is, the number of search locations. By increasing the SNR threshold to a value where only one false positive per search is expected on average, almost all search locations that do not contain matching signal are excluded. The high SNR threshold used in our 50S template search, therefore, limits the template bias in the reconstruction. Indeed, apart from the large additional density corresponding to the 30S subunit, there are also several smaller differences between the

template and the reconstruction within the region of the 50S subunit that can be related to conformational changes occurring during translation (*Figure 3c–d*, see Results). This confirms that template bias in this reconstruction must be small relative to the unbiased signal represented by the reconstruction.

## Template optimization

In this study, we found that applying a B-factor of 85 $\text{Å}^2$ to the 50S template maximized the mean detection SNR in this dataset (*Figure 3—figure supplement 1a*). This value matches closely the B-factor of 86 $\text{Å}^2$ used to correct the in situ reconstruction of the *M. pneumoniae* 70S from subtomograms (*Tegunov et al., 2021*). Since this B-factor affects all atoms in the template equally, we expect that a more accurate method of generating templates that uses variable B-factors to account for local differences in mobility will further improve detection. We propose that given a sufficiently accurate method to calculate template densities, 2DTM could be used to investigate the relative impact of these and other, not yet considered, factors. For example, we could use biochemical restraints to control conformational/compositional heterogeneity and thereby investigate the process of radiation damage which likely varies based on a complex's chemical composition and environment. Alternatively, if we can correctly model sample motion and radiation damage, we could use 2DTM to probe for more detailed aspects of an individual target's structural identity and composition, which may vary based on location in the cell or lifecycle of the organism to be investigated. It will therefore be important in the further development of 2DTM to also develop more reliable and accurate methods to generate templates from atomic coordinates, and to comprehensively model cryo-EM images.

Templates in the present study used density maps generated by *pdb2mrc* (*Tang et al., 2007*), and projections were calculated using the simple linear interpolation algorithms implemented in *cis*-TEM (*Grant et al., 2018*). These simplifications contributed to the acceleration of our 2DTM implementation, compared to the more accurate density modeling and projection calculation used previously (*Rickgauer et al., 2020*; *Rickgauer et al., 2017*). The simplifications, as well as noticeable blurring in some of our images due to deformation of the *M. pneumoniae* cells under the electron beam (*Tegunov et al., 2021*), likely affect the SNR values obtained in our template searches. Indeed, the SNR values we observe are lower on average than expected based on the molecular mass of our template (about 1.2 MDa), as well as previously observed values obtained with different samples and template structures (*Rickgauer et al., 2020*; *Rickgauer et al., 2017*). Furthermore, the *M. pneumoniae* 50S atomic model built into a 3.5 Å map (*Tegunov et al., 2021*) may contain atomic coordinate errors that are larger than those in the 60S model used by *Rickgauer et al., 2020*, which was built into a 2.9 Å map. Coordinate errors will further decrease the SNR values obtained in a template search. Accurate atomic models and methods to simulate realistic cryo-EM images (e.g. *Himes and Grigorieff, 2021*) will therefore be important to maximize the detectability of molecules by 2DTM.

## 2DTM with multiple templates

2DTM employs the matched filter, which is the statistically optimal detector for a deterministic signal in wide-sense stationary noise (*McDonough and Whalen, 1995*). With zero-mean Gaussian noise, the output of the matched filter is an SNR determined by the ratio of the matching signal cross-correlation coefficient to the standard deviation of correlation coefficients in areas of the image devoid of matching signal. Given that the noise is roughly constant for a given image, the SNR measured in 2DTM is ultimately limited by the template's molecular mass, making smaller complexes more difficult to detect. In our earlier work, we limited the SNR threshold to a value allowing, on average, one false positive per search. This threshold is, however, not optimal for every experiment and should be adjusted based on the experimental rationale and design.

A match in 2DTM indicates that the template is sufficiently similar to the target to yield an SNR above the significance threshold but does not indicate that the target conforms to the exact structure of the template. Indeed, significant peaks were detected using the *B. subtilis* 50S as a template which has substantial differences to the *M. pneumoniae* 50S (discussed below). Moreover, since target detection depends on pre-existing structures, detection by 2DTM is necessarily biased towards targets that are sufficiently similar to the state represented by the template, and other states may be missed. As a consequence, targets detected with 2DTM likely reflect a subset of the total complement of in vivo states. To reduce bias, variable regions could be removed from the template,

thereby also reducing the potentially detectable signal in a template search. To detect targets that are in different states without loss of signal, multiple templates representing different states might be used. The problem of mismatched states is particularly evident when performing a local search for bound 30S subunits using the coordinates of detected 50S subunits. Despite the 30S subunits having a molecular mass of ~700 kDa, well above the 300–400 kDa limit expected to be detectable by 2DTM in images of samples with a thickness of 150–200 nm (*Rickgauer et al., 2020*; *Rickgauer et al., 2017*), we found only ~45% of the identified 50S bound with 30S subunits, just over half of the expected 70% based on the analysis of tomograms from *M. pneumonia* (*O'Reilly et al., 2020*, *Figure 6b*). It is unlikely that this low detection rate is due to a bias of 2DTM toward detecting isolated 50S subunits because (i) a 3D reconstruction based on detected 50S subunits shows clear density for the 30S subunit (*Figure 3d*) and (ii) there was no significant difference in the detection of 50S and 70S targets, as validated by 3DTM (*Figure 6—figure supplement 1*). Since the 30S subunit is highly dynamic and can adopt multiple conformations, it is likely that our false negatives arose from conformations that are not sufficiently similar to the 30S template that we used in our search.

Below an SNR of 7.2, the number of detected 2DTM targets without a 3DTM match is less than statistically expected (*Figure 5g*). This suggests that the statistics, which are based on a Gaussian noise model, might suffer from bias, for example because the correlation coefficients calculated for each search location are not completely independent from each other. While further investigation will be needed to model the noise in correlation maps more accurately, the current bias due to residual correlations between search locations results in a slight overestimation of the number of false positives at a given SNR. Incorporating more accurate noise models, and properly accounting for uncertainty in the model (reference structure) will require the replacement of the SNR values currently used to evaluate 2DTM results by a more general probabilistic framework. The use of likelihood ratios, for example, will make 2DTM more robust, as well as open up new avenues to explore molecular heterogeneity.

## 2DTM as a tool to investigate diverse species

One of the limitations of 2DTM is a reliance on pre-existing high-resolution structures. Outside of a few model organisms, high-resolution structures are not available for the vast majority of species. We show that, despite differences in their structures, a *B. subtilis* 50S template can detect 50S in *M. pneumoniae* cells (*Figure 4*). Thus, it is possible to use structures from related organisms to identify the locations and orientations of complexes with 2DTM. Mismatches between the template and cellular target resulting from species-specific structures reduce the 2DTM SNR (*Figure 4c*). By comparing the 2DTM SNRs of a series of structures from different species, it may be possible to infer evolutionary relationships in a manner analogous to DNA sequence comparison. Structural comparison with 2DTM would present the additional advantage of defining structural conservation, which may not be evident by sequence comparison alone, without the need to build a detailed molecular model of homologs in each species.

We have shown that it is possible to generate in situ 3D reconstructions of ribosomes from 2D images of cells, without the need to collect a tilt series (*Figures 3* and *4*). Template bias (discussed above) becomes especially pertinent when using a template from a different species when there is no existing structure to validate the obtained map. In a 3D reconstruction from 50S located by 2DTM with a *B. subtilis* 50S template we identified density corresponding to *M. pneumoniae*-specific protein structures, and failed to detect density corresponding to a *B. subtilis*-specific protein (*Figure 4f–g*). This demonstrates that by using a sufficiently high threshold to prevent a preponderance of false positives, targets identified using 2DTM templates from different species can be averaged to generate in situ 3D reconstructions with minimal template bias. Many biologically important organisms are difficult to grow at scales necessary for protein purification and in vitro structure determination. Cryo-ET is also currently limited by computationally demanding image processing, and is thus low throughput for routine in situ structure determination. 2DTM could be used as an alternative tool to accelerate in situ structure determination in non-model organisms.

## 2DTM and 3DTM complement each other

A significant challenge in using cryo-EM to study complexes in situ arises from the thick and crowded nature of the sample. In tomograms, the background generated by neighboring and overlapping molecules can be partially 'disentangled' from the search targets by resolving all spatial dimensions. This provides an advantage of 3DTM over 2DTM when detecting targets that can be clearly separated in 3D. However, while overlapping density cannot be removed in single-tilt images used for 2DTM, overlapping targets can still be separated based on their different x,y coordinates, views, and defocus values (z coordinate). For example, using 2DTM we were able to discern overlapping 50S ribosomes, as validated by comparison with the 3DTM results (*Figure 6f–i*). Thus, separating overlapping particles does not require collecting a tilt series.

The results of 3DTM do not seem to be correlated with sample thickness within the tested range (*Figure 5—figure supplement 2*). It is difficult to tell if this is due to a better overall performance of 3DTM in thicker samples, for example due to the separation of overlapping densities, or if 3DTM is simply less sensitive to the signal degradation associated with thicker samples, for example by multiple and inelastic scattering. The latter seems more likely since 3DTM depends primarily on low-resolution signal, which in cryo-EM data is usually substantially stronger than high-resolution signal and therefore remains detectable in thicker samples despite signal degradation. While the stronger reliance on low-resolution features makes 3DTM less sensitive to sample thickness and image-degrading factors, its high false positive rate requires extensive expert curation, often exceeding the computational run time of the original search. Even where time is not a concern, manual curation is difficult if not impossible in dense regions of a cell, like the nucleus, or for particles smaller than a ribosome that are not easily visually discernible. Particle classification approaches such as those implemented in *RELION* can be useful for removing false positives. However, their performance on noisy subtomograms of relatively small or rare complexes remains problematic.

Using the full electron dose in a single exposure allows for the inclusion of high-resolution information in a 2DTM search, which improves its precision and enables detection of rare complexes in dense molecular environments, thereby alleviating some of the major constraints of 3DTM. However, as we show, the higher precision of 2DTM comes at the expense of lower sensitivity in thick samples. The lack of multiple sample tilts makes larger features such as membranes more difficult to map in 3D and lowers the positional accuracy perpendicular to the image plane. In principle, 3DTM could also take advantage of high-resolution information, but in practice this is currently not achieved, possibly due to increased specimen motion on multiple exposures, errors in the CTF determination from very low-dose exposures, difficulty in aligning tilt series, and increasing sample thickness as a function of tilt (*Voortman et al., 2014*). New approaches that take into account sample deformation (*Himes and Zhang, 2018*; *Tegunov et al., 2021*) may help boost the high-resolution signal in tomograms.

Better overall performance of template matching may be achieved by combining the high precision of 2DTM with the high sensitivity of 3DTM and broader cellular context in tomograms, making this approach an effective and informative strategy for visual proteomics. In such a strategy, a zero-tilt image with higher dose is collected before collecting a tilt series. The molecules of interest are then identified in both 2D image and 3D tomogram, the 2D search is used to validate hits in the tomogram, and the 3D coordinates provide cellular context in 3D space and higher accuracy in the z coordinates (e.g. *Figure 5a–b*). Other approaches that leverage the different information available from 2D images and 3D tomograms have been suggested previously (*Bartesaghi et al., 2012*; *Sanchez et al., 2020*). Moreover, since the image modalities (2D vs. 3D) are distinct in the two methods, the noise and background in the data are only partially related: (i) Layers of the sample overlapping in the 2D image can be separated in the tomogram. (ii) The imaging parameters (effective image contrast/defocus, electron dose) as well as the random noise accompanying all cryo-EM data differ between the 2D and 3D data. (iii) Template matching in 2D and 3D relies on different resolution ranges – 3DTM depends strongly on shape information (~20 Å) while 2DTM depends on 3–5 Å resolution. Overlapping hits can therefore be considered to be true positives with high confidence. The requirement of dual detection may allow the detection threshold to be lowered in each search, increasing the overall number of true positives. Such an approach could speed up workflows by avoiding the need for labor-intensive classification and refinement of 3DTM datasets, while incorporating the additional structural context provided by a tomogram. We anticipate that a combined

2DTM-3DTM approach could extend high-confidence complex detection to rare molecules in thicker samples.

## Future of template matching and conclusions

We describe here the implementation of a GPU-accelerated 2DTM method into the graphical user interface of the open-source software *cis*TEM. 2DTM effectively detects ribosomes in 2D images of frozen *M. pneumoniae* cells at higher precision, but lower sensitivity than 3DTM in samples thicker than 100 nm. The significance threshold in 2DTM prevents the need for additional manual curation, refinement, averaging or classification, substantially streamlining the process. We propose that an effective, high-confidence strategy for in situ visual proteomics would combine 2DTM and 3DTM and would not be limited to species for which high-resolution structures are available. We also propose that the search space of 2DTM could be further expanded to include a more detailed analysis of the signal present in an image, allowing for interrogation of conformational or compositional variability, by searching with a multi-template library. We demonstrated that 3D reconstructions from targets detected by 2DTM reveal new features not present in the template. This technique, therefore, has the potential to deliver in situ structures at high resolution, similar to subtomogram averaging but requiring less experience and time. While it requires prior knowledge of the target structure to be reconstructed, that is, a template, the 2DTM workflow (including data collection) is significantly faster than tomography and may therefore be more suitable for high-throughput studies. Finally, the need for averaging limits detailed interrogation of molecular structures to more abundant complexes. The spatial organization, structures, composition, and functional states of rare complexes can still be studied in cells by 2DTM, provided they are detectable in 2D images. With further improvements, we expect 2DTM to reveal new insights into the molecular mechanisms of biological processes in their native cellular context.

# Materials and methods

### Key resources table

| Reagent type (species) or resource | Designation | Source or reference | Identifiers | Additional information |
|---|---|---|---|---|
| Cell line (*Mycoplasma pneumonia*) | M129 | **O'Reilly et al., 2020** | ATCC 29342 | |
| Software, algorithm | *cis*TEM | This paper and Grant **Grant et al., 2018**. | doi:10.5281/zenodo.4603401 | https://cistem.org/ |

## Cell culture, grid preparation, and cryo-EM imaging

*Mycoplasma pneumoniae* sample preparation and cryo-EM imaging were carried out as described previously (*O'Reilly et al., 2020*; *Tichelaar et al., 2020*). In brief, *M. pneumoniae* cells were grown on Quantifoil gold grids in modified Hayflick medium and the grid was quickly washed with PBS buffer with 10 nm gold fiducial beads (Aurion, Germany) before plunge-freezing. The grids were imaged in a 300 keV Krios TEM (ThermoFisher) equipped with a direct detector (Gatan) and a quantum post-column energy filter (Gatan). Data collection was done using SerialEM (*Mastronarde, 2005*).

For the 2D images paired with tilt series data, 19 cells were first imaged with parameters commonly used for single-particle cryo-EM, with a magnification of 215,000 (pixel size 0.65 Å), exposure time of 2 s (20 frames, total exposure 32 e$^-$/Å$^2$), and a target defocus between 0.2 μm and 0.5 μm. Tilt series were collected using the dose-symmetric scheme (*Hagen et al., 2017*) with the following settings: magnification 81,000 (pixel size 1.7 Å), tilt range −60° to 60° with 3° increment, 1 s exposure per tilt, and a total exposure of ~129 e$^-$/Å$^2$. The target defocus remained the same within tilt series, and ranged from 1.5 μm to 3.5 μm between tilt series.

A dataset of 220 2D images of *M. pneumoniae* was collected on a K3 camera (Gatan) and at a magnification of 81,000 (pixel size 1.053 Å). The K3 camera was run in non-CDS mode with the following settings: exposure time 1.678 s, 24 frames, total exposure ~31 e$^-$/Å$^2$.

## 2D template generation

To generate templates for 2DTM, we used the computer program *pdb2mrc* (*Tang et al., 2007*) to convert PDB-formatted atomic coordinates into 3D density. To reduce potential aliasing, we initially generated an over-sampled density map at half the pixel size of the image to be searched. The sampling rate was then halved to the final pixel size by Fourier cropping using the program *resample* from the *cis*TEM software package (*Grant et al., 2018*). The resulting 3D density was low-pass filtered with the standalone program *bfactor* (*Grigorieff, 2021b*) using a B-factor of 85 Å$^2$ and placed into a $256 \times 256 \times 256$ voxel box, about twice the size of the 50S ribosomal subunit it contained. The final pixel size for searching the 19-image dataset of *M. pneumoniae* that was compared with tomograms was 1.27 Å. For computational efficiency, we used a final pixel size of 1.5 Å to search the 220-image dataset. As template models, we used the *M. pneumoniae* ribosome (PDB: 7OOC and 7OOD) and the *B. subtilis* ribosome (PDB: 3J9W). We separated the small and large ribosomal subunits and, for the *B. subtilis* ribosome, removed the mRNA, tRNAs and MifM protein from the coordinate file using custom Perl scripts. The resulting coordinates were aligned with *USCF Chimera* (*Pettersen et al., 2004*) to the 50S template used for 3DTM to place them in a common coordinate system. This ensured that the 3D positions and Euler angles found by 2DTM and 3DTM referred to the same coordinate system and could be directly compared. For local 30S searches, the atomic coordinates corresponding to the body of the *M. pneumoniae* 30S were used to generate a template as described above. For this we combined the results of searches with two templates, one with the proteins and rRNA sequence corresponding to the 30S head and one without, to account for the greater conformational variability of the small subunit relative to the large subunit.

## 2D template matching in *cis*TEM

Movie frames were aligned with *unblur* using the 'optimal exposure filter' (*Grant and Grigorieff, 2015*) and including all frames in the final sum, unless otherwise indicated. Defocus was determined by *CTFFIND4* (*Rohou and Grigorieff, 2015*) from within the *cis*TEM GUI (*Grant et al., 2018*). Templates for 2DTM were imported into *cis*TEM as 3D volumes and 2DTM was performed on all images, including a defocus search over a 2400 Å range centered on the average defocus determined by *CTFFIND4* with a 200 Å step, using a 2.5° out-of-plane angular search step and a 1.5° in-plane angular search step, assuming C1 symmetry and a minimum target - target distance (peak radius) of 10 pixels (~13 Å). As previously described, the SNR values resulting from 2DTM were further normalized by subtracting the mean of the SNR values for all orientations at each location, and dividing by their standard deviation (*Rickgauer et al., 2020*; *Rickgauer et al., 2017*). The best peak radius to use in a search depends on the B-factor affecting the signal in the images and may, therefore, vary for different experiments.

## Detection of 30S by local search using 50S 2DTM coordinates

To detect 30S subunits bound to 50S subunits, we wrote *refine_template*, a computer program that is accessible through *cis*TEM's GUI, or as a command line tool. *refine_template* reads the 3D template and the output files of a template search, including the MIP and alignment parameters for detected targets. Using these data, it calculates template projections and performs a local refinement of the projection parameters. This refinement can be used, for example, to find the defocus values that maximize the cross-correlation coefficient between projection and image. It can also be used to detect molecules and complexes bound to already detected targets. In this case, *refine_template* is used with the search results obtained for the detected targets, but with the bound complex to be detected as the input template. The locally refined alignment parameters found for the bound complex will be close to those found for the original template, provided both the original template used in the search and the bound complex in the refinement share the same coordinate system.

To ensure the coordinate systems of the *M. pneumoniae* 30S and 50S subunits were aligned, the coordinates for the 30S and 50S subunits were separated from the 70S into two coordinate files and used to generate 3D templates for 2DTM for each of them (see above). For this last step, it was important not to use the *-center* flag of *pdb2mrc* to make sure the 3D densities remained in their correct locations relative to the other subunit. 50S targets were then detected using the 50S template, followed by local refinement with *refine_template* and the 30S template. For the refinement,

we searched within an angular range of ±12.5˚ using a 2.5˚ step. A 30S subunit was deemed to be present when the refined x,y coordinates were within 20 Å of the original coordinates found for the 50S template.

## 3D reconstruction using 50S 2DTM coordinates

We wrote a new computer program called *prepare_stack_matchtemplate*, as part of the *cis*TEM image processing package. It reads the results of a template search, including the MIP with the peaks indicating detected targets, as well as the Euler angles of the detected targets, the original image that was searched, and other imaging parameters (defocus, amplitude contrast, lens aberrations, beam energy, and pixel size). On output, it generates a stack of boxed-out targets and a list of alignment parameters that can be used for 3D reconstruction with *cis*TEM. Using *prepare_stack_matchtemplate* and *cis*TEM, we generated a reconstructions from 5080 and 1172 50S targets detected in 220 images of *M. pneumoniae* cells, using the Euler angles and x,y locations assigned by the *M. pneumoniae* and *B. subtilis* 50S template search, respectively. The reconstructions had nominal resolutions of 4.3 Å and 5.5 Å, respectively (FSC threshold of 0.143, *Rosenthal and Henderson, 2003*; *Figure 3—figure supplement 1e*, *Figure 4—figure supplement 1b*). The masked FSC curves between the *M. pneumoniae* 30S structure and the 3D reconstructions from targets found with the *M. pneumoniae* and *B. subtilis* 50S templates revealed interpretable information out to 15 Å and 20 Å resolution, respectively (*Figure 4—figure supplement 1d*). We therefore low-pass filtered the reconstructions to the common 20 Å resolution.

## 3D template matching

For tilt series data, movie frames were aligned on-the-fly using the SerialEM plug-in *alignframes*. Tilt series alignment was carried out in IMOD (*Mastronarde and Held, 2017*). Warp was used to estimate CTF, import tilt series alignments from IMOD, reconstruct tomograms and subtomograms (*Tegunov et al., 2021*). Cell thickness was estimated based on central y,z sections of the tomograms.

We used PyTom to perform 3DTM of tomograms as described previously (*Hrabe et al., 2012*; *O'Reilly et al., 2020*), binning the tomograms to a pixel size of 6.8 Å and using an angular sampling step of 19.95˚. The 50S ribosome template was generated from a previous 50S class average from *M. pneumoniae* (*O'Reilly et al., 2020*). The density was low-pass filtered to 30 Å resolution prior to template matching in PyTom. 3DTM hits were extracted from the scores map generated by PyTom and were ranked by cross-correlation scores for each tomogram. Subtomograms for the top 600 hits in each tomogram were reconstructed using Warp. 3D classification and refinement were done in *RELION* 3.0.8 (*Zivanov et al., 2018*). In total, 3062 70S and 6336 50S subtomograms were classified, after removing 1827 false positives, that is, particles that could not be aligned (*Figure 5—figure supplement 1*).

## Comparison of 2DTM and 3DTM

To compare targets found by 2DTM and 3DTM, we wrote a new computer program, *align_coordinates*, as part of *cis*TEM (*Grant et al., 2018*), to identify and align target coordinates. *align_coordinates* assumes that the z axes of the coordinate systems of the 2DTM and 3DTM searches are aligned, that is, they refer to the same sample tilt. Therefore, to find the angular alignment and x,y offset between a set of reference coordinates and a second set of sample coordinates, only the x,y coordinates need to be considered. In a first step, *align_coordinates* calculates all possible vectors between any two x,y coordinates in a coordinate set. A rotation matrix is applied to the sample coordinate vectors, and the vector with the smallest vector difference within a given tolerance is found with respect to each reference vector. The algorithm performs a systematic search of the rotation angle in 0.5˚ steps to find the rotation that produces the largest number of matching vectors while giving a higher weight to better matching vectors. The result of this systematic search is a rough rotation angle and a list of corresponding coordinates that define a coordinate transform. The search also identifies coordinates in each set that do not have corresponding coordinates in the other set. The coordinate transform is then refined, and the list of corresponding coordinates is updated using a finer local search with angular step of 0.01˚ and 1 Å steps along the x and y axes.

The best coordinate transform is selected based on a least-squares fit between the corresponding coordinates.

In the comparison of 2DTM and 3DTM coordinates, the latter were used as a reference set, and the distance threshold for correspondence was set to 110 Å. The final list of corresponding targets was further reduced by limiting distances to 100 Å and below. By providing the dimensions of the field of view of the images used for 2DTM, it was also possible to identify the corresponding area within the larger field of view of tomograms analyzed by 3DTM. For the calculation of percentages of corresponding targets identified by each search, only targets in the overlapping areas were counted.

To compare the angular orientations for a given coordinate pair, the 2DTM template was first aligned relative to the 3DTM template (see above), and the Euler angles of the detected targets were recorded. The total angular distance between each orientation, represented by the rotation matrices $R_2$ and $R_3$, given by the Euler angles, was calculated using the equation (*Huynh, 2009*)

$$\vartheta_{2,3} = \cos^{-1}\left[\frac{tr\left(R_2 R_3^T\right) - 1}{2}\right]$$

(1)

## Acknowledgements

We are grateful to Wim Hagen for advice on cryo-EM data acquisition, and Peter Rickgauer for providing numerous fruitful discussions and advice on implementing template matching in *cis*TEM. JM acknowledges support from the EMBL and the European Research Council (760067).

## Additional information

### Competing interests
Nikolaus Grigorieff: Reviewing editor, *eLife*. The other authors declare that no competing interests exist.

### Funding

| Funder | Grant reference number | Author |
| --- | --- | --- |
| European Research Council | 760067 | Julia Mahamid |
| Howard Hughes Medical Institute | | Bronwyn A Lucas<br>Benjamin A Himes<br>Nikolaus Grigorieff |

The funders had no role in study design, data collection and interpretation, or the decision to submit the work for publication.

### Author contributions
Bronwyn A Lucas, Conceptualization, Data curation, Formal analysis, Validation, Investigation, Visualization, Methodology, Writing - original draft, Writing - review and editing; Benjamin A Himes, Conceptualization, Software, Formal analysis, Validation, Investigation, Visualization, Methodology, Writing - original draft, Writing - review and editing; Liang Xue, Conceptualization, Data curation, Formal analysis, Validation, Visualization, Writing - review and editing; Timothy Grant, Conceptualization, Software, Visualization, Methodology, Writing - review and editing; Julia Mahamid, Conceptualization, Resources, Supervision, Funding acquisition, Validation, Project administration, Writing - review and editing; Nikolaus Grigorieff, Conceptualization, Resources, Data curation, Software, Formal analysis, Supervision, Funding acquisition, Validation, Investigation, Methodology, Writing - original draft, Project administration, Writing - review and editing

### Author ORCIDs

Benjamin A Himes (iD) http://orcid.org/0000-0001-7777-0298
Liang Xue (iD) http://orcid.org/0000-0003-4368-2526
Timothy Grant (iD) https://orcid.org/0000-0002-4855-8703

Julia Mahamid  http://orcid.org/0000-0001-6968-041X
Nikolaus Grigorieff  https://orcid.org/0000-0002-1506-909X

**Decision letter and Author response**
Decision letter https://doi.org/10.7554/eLife.68946.sa1
Author response https://doi.org/10.7554/eLife.68946.sa2

## Additional files

### Supplementary files
• Transparent reporting form

### Data availability

All the code used for the 2D template matching has an open source license and is freely available from the cisTEM github repository, https://github.com/timothygrant80/cisTEM. A snapshot of the code used for this study is available on Zenodo, https://doi.org/10.5281/zenodo.4603401. The images of M. pneumoniae analyzed for this work, as well as the 70S reconstructions have been deposited in the EMPIAR and EMDB databases, https://www.ebi.ac.uk/pdbe/emdb/empiar/ and https://www.ebi.ac.uk/pdbe/emdb.

The following datasets were generated:

| Author(s) | Year | Dataset title | Dataset URL | Database and Identifier |
|---|---|---|---|---|
| Lucas BA, Himes BA, Xue L, Grant T, Mahamid J, Grigorieff N | 2021 | 3D reconstruction generated using the locations and orientations of 1172 50S subunits detected in 220 image | https://www.ebi.ac.uk/pdbe/entry/emdb/EMD-23772 | Electron Microscopy Data Bank, EMD-23772 |
| Lucas BA, Himes BA, Xue L, Grant T, Mahamid J, Grigorieff N | 2021 | 3D reconstruction generated using the locations and orientations of 5080 50S subunits detected in 220 image | https://www.ebi.ac.uk/pdbe/entry/emdb/EMD-23771 | Electron Microscopy Data Bank, EMD-23771 |
| Lucas BA, Himes BA, Xue L, Grant T, Mahamid J, Grigorieff N | 2021 | Locating Macromolecular Assemblies in Cells by 2D Template Matching with cisTEM | https://www.ebi.ac.uk/pdbe/emdb/empiar/entry/10731/ | EMPIAR, 10.6019/EMPIAR-10731 |
| Lucas BA, Himes BA, Xue L, Grant T, Mahamid J, Grigorieff N | 2021 | Locating Macromolecular Assemblies in Cells by 2D Template Matching with cisTEM | https://www.ebi.ac.uk/pdbe/emdb/empiar/entry/10727/ | EMPIAR, 10.6019/EMPIAR-10727 |

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
