## [Decision Letter]

**Acceptance summary:**

You have fully addressed the concerns of the reviewers. While in some cases you have explained that resolving certain issues is beyond the scope of the present paper, the manuscript as it currently exists will be a useful contribution to the area of template matching for locating complexes in cells.

**Decision letter after peer review:**

Thank you for submitting your article "Locating Macromolecular Assemblies in Cells by 2D Template Matching with cisTEM" for consideration by *eLife*. Your article has been reviewed by 3 peer reviewers, including Edward H Egelman as the Reviewing Editor and Reviewer #1, and the evaluation has been overseen by Richard Aldrich as the Senior Editor. The following individual involved in review of your submission has agreed to reveal their identity: Elizabeth Villa (Reviewer #2).

Essential revisions:

1) A point of concern is the degree of reference-bias in the results of the 2DTM approach. The authors acknowledge this concern and that conventional use of the FSC is not a suitable validation metric for this approach nor for determining an appropriate filtering cutoff for a resulting reconstruction. The proposed validation metric of the emergence of additional known density features in a reconstruction, which are not present in the template, resulting from 2DTM hits is sensible. However, emergence of additional unknown densities in a reconstruction resulting from cellular data will be difficult to segregate from noise, especially since filtering of the reconstruction is determined ad hoc instead of by an objective metric.

2) The work remains on an empirical level as surprising advantages of the 2D approach compared to 3D are revealed, but there is little effort to get to the basis of these observations. Moreover, details on the compared 3D approach (and its parameter optimization analogous to the 2D approach), which the rather general conclusions would require, are missing. Lastly, the 3D approach has been applied to the strongly pre-irradiated sample, which may make observations such as a lower specificity in the 3D case almost a self-fulfilling prophecy. Thus, the 2D vs 3D comparison is not convincing in the current form. The 2D implementation of in situ structure determination is interesting and of potential interest to a large audience. However, the comparison to the 3D equivalent appears somewhat incomplete and the rather general conclusions require further validation. In the absence of further validation, the conclusions need to be toned down.

3) Which concrete recommendations do the authors provide with regards to data acquisition? The researcher has to make a decision whether to spend the most valuable first ~30 e-/Å^2^ on a projection or a tilt series. In this manuscript, the authors argue for the former, while Mahamid and colleagues have obtained beyond 4 Å resolution using the latter strategy (Tegunov et al., 2021). This leaves people somewhat puzzled.

4) In this work, the provided FSCs suggest that resolutions in the range of 5 Å are achieved, but all figures are low pass filtered to ~20 Å, which appears odd. Of note, the FSCs do not exhibit an extended plateau with FSC approx. 1 at low frequencies, which is normally observed. Please explain.

5) Cross-resolutions of reconstructions and 30S (from B. subtilis and M. pneumoniae) should be provided as an assessment.

6) L. 298: "The pre-exposure of 32 e-/Å^2^ affects the signal in the tomograms at higher resolution but is expected to have only a small effect in the resolution range relevant for 3DTM, i.e., 20 Å and lower (Grant and Grigorieff, 2015)." This is a very key assumption of the entire study that must be validated experimentally. Nucleic acids are known to considerably more beam sensitive than amino acids and hence 32 e-/Å^2^ is typically considered a very high dose for ribosomes. The authors must do the reverse experiment to validate the stated assumption, which is to spend the first 32 e-/Å^2^ on a tilt series, then acquire the 32 e-/Å^2^ 2D image, and finally resume the tilt series.

7) For 3D template matching essential computational details are omitted. Importantly, the spatial and rotational sampling (i.e., voxel size and rotational increment/number of rotated templates) is not specified. Given that the authors point out the importance of rotational sampling this omission is surprising.

8) Regarding the above remark: what is the authors' expectation if sampling is increased for the 3D case (and first 30 e/Å^2^ on the tilt series): will the suggested trends on 3D vs 2D approach still hold? In other words: is (limited) computation the problem in the 3D case or is there a conceptual advantage resulting in higher specificity for the 2D approach?

9) To make general statements about 2D vs 3D template matching the authors must ensure that they use comparable sampling – or use some other normalization of their approaches (e.g. computation time).

10) The paper lacks any discussion of the 2D vs 3D results in the light of the dose fractionation theorem, as stated by Hegerl and Hoppe theoretically and experimentally confirmed by McEwen. This theorem states that the information content of 2D image or 3D volume imaged with the same dose is the same, albeit different structure factors are sampled in Fourier space.

---

## [Author Response]

Essential revisions:1) A point of concern is the degree of reference-bias in the results of the 2DTM approach. The authors acknowledge this concern and that conventional use of the FSC is not a suitable validation metric for this approach nor for determining an appropriate filtering cutoff for a resulting reconstruction. The proposed validation metric of the emergence of additional known density features in a reconstruction, which are not present in the template, resulting from 2DTM hits is sensible. However, emergence of additional unknown densities in a reconstruction resulting from cellular data will be difficult to segregate from noise, especially since filtering of the reconstruction is determined ad hoc instead of by an objective metric.

Validation of unknown features that do not reach a resolution where secondary or primary structural features can be recognized is difficult, as decades of “blobology” preceding the cryoEM resolution revolution have shown. However, in the case of template bias with no additional refinement, the emergence of any additional features above the noise level cannot arise only from noise and will therefore reliably indicate true additional density in the detected targets at that location. Nevertheless, we have added an evaluation of the 3D reconstructions (lines 230 – 232, lines 764 – 768, Figure 4—figure supplement 1d) obtained from targets detected by 2DTM by local cross-FSC with the *M. pneumoniae* 30S template.

Another issue that the reviewers may have had in mind is that the additional density may represent a mixture of structures and/or conformations (heterogeneity) and therefore cannot easily be interpreted without further classification. 3D reconstruction and classification to assess structural heterogeneity by will be the subject of a future study.

2) The work remains on an empirical level as surprising advantages of the 2D approach compared to 3D are revealed, but there is little effort to get to the basis of these observations. Moreover, details on the compared 3D approach (and its parameter optimization analogous to the 2D approach), which the rather general conclusions would require, are missing. Lastly, the 3D approach has been applied to the strongly pre-irradiated sample, which may make observations such as a lower specificity in the 3D case almost a self-fulfilling prophecy. Thus, the 2D vs 3D comparison is not convincing in the current form. The 2D implementation of in situ structure determination is interesting and of potential interest to a large audience. However, the comparison to the 3D equivalent appears somewhat incomplete and the rather general conclusions require further validation. In the absence of further validation, the conclusions need to be toned down.

We would like to reiterate that the purpose of our work is to evaluate target detection in cells by 2DTM. We do not suggest replacing 3DTM with 2DTM, nor do we suggest that calculating 3D reconstructions from 2DTM targets is superior to reconstructions obtained through subtomogram averaging. Both approaches have their different strengths and weaknesses, as described in our manuscript, and therefore, they are complementary to each other. We agree with the reviewers that it is helpful to recapitulate the parameters used for 3DTM. Our 3DTM protocol follows standard procedures that are well-established in the field of cryo-tomography and optimize the identification of targets. As mentioned in our manuscript and described by Hrabe et al. (2012), we used PyTom to search our tomograms. The pixel size in the tomograms was 6.8 Å, and 3DTM used angular sampling with points separated by 19.95°, which was previously shown to produce good results for ribosomes. We have now added these details to the Methods section of our manuscript.

Regarding pre-irradiation of the sample for 2DTM, we understand the reviewers’ concern that this may affect 3DTM of the subsequently recorded tomograms. However, the effective B-factor of the subtomograms estimated by *RELION* was 1700 Å2 for non-pre-irradiated tomograms, as determined for data processed in O’Reilly et al. (2020), and 2000 Å2 for pre-irradiated tomograms collected for the present study. At 30 Å, the resolution of the template commonly used for 3DTM (see Methods), these B-factors will attenuate the signal by factors of 0.62 and 0.57, respectively, a difference of 8%. Another indication that the pre-exposure has only a small effect on 3DTM is provided in Figure S5 of O’Reilly et al. (2020). This figure shows a plot similar to our plot in Figure 5—figure supplement 1c. Both figures show the rate of false positives of targets detected by 3DTM, based on 3D classification in *RELION*. For example, for the top-scoring targets, both plots indicate a false-positive rate of about 5%. For targets ranked around 200 from the top, about 50% are false positives, and for targets ranked around 400 from the top, the false-positive rate is about 65% in both cases. We have amended our Discussion of the effect of pre-irradiation in our revised manuscript (lines 299 – 301, 311 – 313), citing the comparisons with O’Reilly et al. (2020), and acknowledge that there is a small difference in the signal attenuation between non-pre-irradiated and pre-irradiated samples that may introduce a small bias in our analysis in favor of 2DTM.

3) Which concrete recommendations do the authors provide with regards to data acquisition? The researcher has to make a decision whether to spend the most valuable first ~30 e-/Å^2^ on a projection or a tilt series. In this manuscript, the authors argue for the former, while Mahamid and colleagues have obtained beyond 4 Å resolution using the latter strategy (Tegunov et al., 2021). This leaves people somewhat puzzled.

In the present study, our goal is to investigate the potential of 3D reconstruction from 2DTM targets, in analogy to subtomogram averaging, to show that additional features can be visualized in detected targets that are not present in the template, despite the well-known problem of template bias. We do not envision that 2DTM will replace de novo 3D reconstruction of whole complexes by subtomogram averaging since 2DTM, by design, requires a large part of the structure to be known at high resolution to find 2D targets. A full investigation of how to optimize 3D reconstruction from detected 2DTM targets, including recommendations for how to collect data for this purpose, will be done in a future study. We hope that the 2DTM approach will be useful in projects where at least part of the 3D structure is known, and where their localization in the cell helps in understanding their biological function. Following future efforts, we hope to develop this approach to dissect their conformational heterogeneity.

4) In this work, the provided FSCs suggest that resolutions in the range of 5 Å are achieved, but all figures are low pass filtered to ~20 Å, which appears odd. Of note, the FSCs do not exhibit an extended plateau with FSC approx. 1 at low frequencies, which is normally observed. Please explain.

We agree with the reviewers that additional justification for the 20-Å low-pass filtering of the reconstructions will improve the manuscript. We have evaluated the local cross-FSC with the *M. pneumoniae* 30S template since template bias will not affect this part of the reconstruction (see new Figure 4—figure supplement 1d). The FSC = 0.5 threshold, as required for map-to-model comparison, indicates an estimated resolution of ~15 Å for the reconstruction obtained from targets found with the *M. pneumoniae* template, and ~20 Å for the reconstruction obtained from targets found with the *B. subtilis* template. As described in the revised manuscript (lines 230 – 232, lines 764 – 768), we have filtered all reconstructions with a 20-Å low-pass filter for consistency.

Regarding the shapes of the reported FSC curves, the 3D reconstructions we obtained differ in several ways from those commonly reported, and these differences will have an effect on the FSC curves. Compared to most single-particle reconstructions, our images contain cellular background, particle poses were not refined (angles, x-y positions, and defocus), and they contain a mixture of conformations, especially in the 30S region.

5) Cross-resolutions of reconstructions and 30S (from B. subtilis and M. pneumoniae) should be provided as an assessment.

We thank the reviewers for this suggestion. This information is now included (lines 230 – 232, lines 764 – 768). Please see also our reply to comment #4.

6) L. 298: "The pre-exposure of 32 e-/Å^2^ affects the signal in the tomograms at higher resolution but is expected to have only a small effect in the resolution range relevant for 3DTM, i.e., 20 Å and lower (Grant and Grigorieff, 2015)." This is a very key assumption of the entire study that must be validated experimentally. Nucleic acids are known to considerably more beam sensitive than amino acids and hence 32 e-/Å^2^ is typically considered a very high dose for ribosomes. The authors must do the reverse experiment to validate the stated assumption, which is to spend the first 32 e-/Å^2^ on a tilt series, then acquire the 32 e-/Å^2^ 2D image, and finally resume the tilt series.

We agree with the reviewers that this statement requires further justification. Please see our reply to comment #2. We would like to note that reversing the experiment would be uninformative since 2DTM relies on the high-resolution signal much more strongly than 3DTM (see above).

7) For 3D template matching essential computational details are omitted. Importantly, the spatial and rotational sampling (i.e., voxel size and rotational increment/number of rotated templates) is not specified. Given that the authors point out the importance of rotational sampling this omission is surprising.

We agree and now provide these details in the Methods, lines 776 – 778 (see our reply to comment #2).

8) Regarding the above remark: what is the authors' expectation if sampling is increased for the 3D case (and first 30 e/Å^2^ on the tiltseries): will the suggested trends on 3D vs 2D approach still hold? In other words: is (limited) computation the problem in the 3D case or is there a conceptual advantage resulting in higher specificity for the 2D approach?

The tomograms we searched using 3DTM are limited by the inherent B-factor of 1700 – 2000 Å2 (see our reply to comment #2). This means that finer sampling will not increase the number of detected targets. Looking ahead, image processing of tomographic data is likely to improve steadily and we expect that this will result in better preservation of the high-resolution signal in the reconstructed tomograms. In the ideal case, the signal across all resolution ranges is preserved perfectly. Provided a good model for the noise in tomograms can also be developed, 3DTM may be as effective as, or even superior to 2DTM in terms of detection precision and sensitivity. As the reviewers point out, there might be other reasons to prefer 2DTM over 3DTM. A major advantage arises from the time it takes to acquire 2D images, compared to tilt series and their subsequent alignment and reconstruction (not required for 2DTM), in addition to computational efficiency (when using similar spatial and angular sampling rates). We have added these points in our revised Discussion, lines 619 – 624.

9) To make general statements about 2D vs 3D template matching the authors must ensure that they use comparable sampling – or use some other normalization of their approaches (e.g. computation time).

As explained in our replies to comments #2 and #8, we performed 3DTM under well-established conditions, which have been previously optimized to produce optimal detection for each approach.

10) The paper lacks any discussion of the 2D vs 3D results in the light of the dose fractionation theorem, as stated by Hegerl and Hoppe theoretically and experimentally confirmed by McEwen. This theorem states that the information content of 2D image or 3D volume imaged with the same dose is the same, albeit different structure factors are sampled in Fourier space.

The publications by Hegerl and Hoppe (1976), and McEwen et al. (1995) revolve around the visualization of sample features at a given significance (i.e., signal-to-noise ratio) using different dose fractionation schemes, ranging from a single exposure, to tilt series of tens to hundreds of exposures. A primary reason for investigating this question stems from the fact that the distribution of noise in the resulting image or reconstruction depends on the distribution of tilts. This is nicely demonstrated in Figure 3 of McEwen et al. where the slices through the 3D reconstruction (third row) are noticeably low-pass filtered compared to the single-tilt images (second row), both recorded with the same total exposure. As explained in our manuscript (lines 633 – 639), the signal-to-noise profiles of 2D images and tomograms are quite different, for multiple reasons including interpolation and weighting in 3D reconstructions, alignment errors in the tilt series, and tilt-dependent sample thickness and radiation damage. These differences results in different detection characteristics of 2DTM and 3DTM. The definition of detection also differs fundamentally between the various studies. While Hegerl and Hoppe consider the (low-resolution) contrast of a sphere and McEwen et al. use the more qualitative criterion of being able to see features in difference images, we use a matched filter and a significance threshold based on a statistical criterion. Finally, neither Hegerl and Hoppe nor McEwen et al. address the question of detecting molecules against a background of other molecules, and how 3D information can be used to remove some of the overlapping density that is present in 2D images. We therefore think that a discussion of these publications in our manuscript would be confusing and not appropriate to include.